# Evolution of the conformational dynamics of the molecular chaperone Hsp90

Stefan Riedl[1], Ecenaz Bilgen[2], Ganesh Agam [2], Viivi Hirvonen[3], Alexander Jussupow[3], Franziska Tippl[1], Maximilian Riedl [1], Andreas Maier[1], Christian F. W. Becker [4], Ville R. I. Kaila [3], Don C. Lamb[2] & Johannes Buchner [1] ✉

Hsp90 is a molecular chaperone of central importance for protein home-ostasis in the cytosol of eukaryotic cells, with key functional and structural traits conserved from yeast to man. During evolution, Hsp90 has gained additional functional importance, leading to an increased number of inter-acting co-chaperones and client proteins. Here, we show that the overall conformational transitions coupled to the ATPase cycle of Hsp90 are con-served from yeast to humans, but cycle timing as well as the dynamics are significantly altered. In contrast to yeast Hsp90, the human Hsp90 is char-acterized by broad ensembles of conformational states, irrespective of the absence or presence of ATP. The differences in the ATPase rate and con-formational transitions between yeast and human Hsp90 are based on two residues in otherwise conserved structural elements that are involved in trig-gering structural changes in response to ATP binding. The exchange of these two mutations allows swapping of the ATPase rate and of the conformational transitions between human and yeast Hsp90. Our combined results show that Hsp90 evolved to a protein with increased conformational dynamics that populates ensembles of different states with strong preferences for the N-terminally open, client-accepting states.

Molecular chaperones are a highly expressed, ubiquitous class of heat shock proteins (Hsps) that are present in virtually all living organism[1]. As a molecular chaperone, the heat shock protein-90, Hsp90, stabilizes other proteins, termed clients, and ensures their biological function[2–5]. Even under physiological conditions, Hsp90 is engaged in a broad spectrum of essential cellular processes[5–8]. In cancer cells, upregulated Hsp90 maintains the integrity and conformation of overexpressed oncoproteins and activates important clients related to tumor growth[9,10]. Consequently, in the last decades, inhibitors have been developed with a view of modulating Hsp90 activity in cancer therapy and as a targeted treatment for folding-related diseases[9,11–14].

Hsp90 consists of three distinct structural domains with different functionalities. The N-terminal domain (NTD) contains the nucleotide-binding site, which belongs to the GHKL ATPase superfamily (Gyrase, Hsp90, histidine kinase, MutL)[15,16]. The Hsp90 GHKL domain comprises a β-sheet framed by α-helices and a flexible, α-helical lid on top[17]. Several inhibitors, including radicicol and geldanamycin, have been identified to interact with the ATP-binding site with nanomolar affinities[18,19]. The NTD is connected to the middle domain (MD) by a highly flexible, charged linker that affects the conformational ensem-ble and influences the chaperone's function[20–23]. The MD is responsible for substrate recognition and contacts the ATP γ-phosphate via a

[1]Center for Protein Assemblies, Department Bioscience, School of Natural Sciences, Technical University Munich, Garching, Germany. [2]Department of Chemistry and Center for Nanoscience, Ludwig-Maximilians-Universität Munich, Munich, Germany. [3]Department of Biochemistry and Biophysics, The Arrhenius Laboratories for Natural Sciences, Stockholm University, Stockholm, Sweden. [4]Institute of Biological Chemistry, Faculty of Chemistry, University of Vienna, Vienna, Austria. ✉e-mail: johannes.buchner@tum.de

catalytic loop during hydrolysis[24,25]. The carboxy-terminal domain (CTD) is the primary dimerization site of Hsp90 and contains a highly conserved Met-Glu-Glu-Val-Asp (MEEVD) motif at the C-terminal end, which enables the recruitment of co-chaperones containing the tetratricopeptide repeat (TPR) domain[26–29].

During its ATP-dependent chaperone cycle, Hsp90 undergoes large conformational changes required for cycle progression (Fig. 1A)[30,31]. *Apo*-Hsp90 populates an open conformation with distant NTDs[32,33]. After ATP binding, the N-terminal lid closes over the bound nucleotide, exposing the α1-helix to the solvent. This induces dimerization of the NTDs involving swapping of the β1-strands between the

dimers leading to an intermediate closed state (closed state 1)[34–36]. Subsequently, the MD rearranges in a manner that allows the catalytic loop to interact with the ATP γ-phosphate (closed state 2)[25]. The interplay of the catalytic loop's arginine piston and the phosphate has proven to be essential for ATP hydrolysis[24]. The cycle is concluded with the re-opening of the dimer.

In recent years, the ATPase cycle of the yeast Hsp90 (yHsp90) has been well characterized. In contrast, human Hsp90 (hHsp90) is far less well studied and leaves many open questions regarding its intrinsic ATPase activity. Despite shared structural components and a high sequence identity, yeast and human Hsp90 exhibit structural

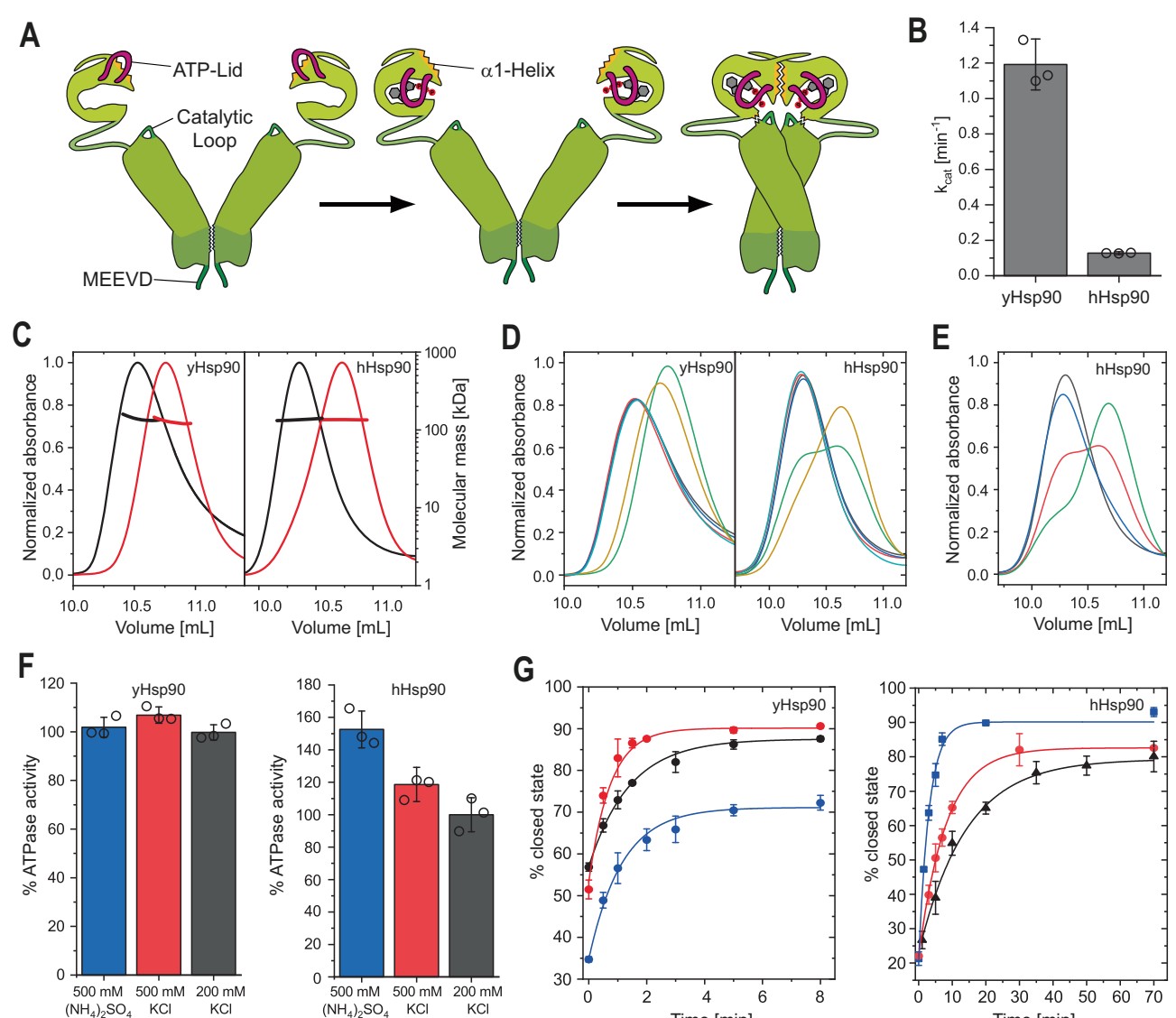

**Fig. 1 | Nucleotide induced closing of yHsp90 and hHsp90. A** Schematic illustration of Hsp90 closing upon nucleotide binding. The N-terminal lid (magenta), the α1-helix (yellow) and the catalytic loop (dark green) are highlighted. **B** ATPase activity of yHsp90 and hHsp90 measured via a regenerative ATPase assay. Data is presented as mean values +/− SD from technical replicates (*n* = 3). **C** Size exclusion chromatography-multi-angle light scattering (SEC-MALS) elution profiles of yHsp90 (left) and hHsp90 (right). Proteins were pre-incubated without (black) or with ATPγS (red) to induce closing. Shifts to higher elution volumes indicate conformational rearrangements to a more compact state. **D** yHsp90 (left) and hHsp90 (right) elution profiles from SEC after incubation (yHsp90: 30 °C, 30 min, hHsp90: 37 °C, 4 h) with varying nucleotides (black: w/o, red: ADP, blue: ATP, green: AMP-PNP, yellow: ATPγS, teal: ADP-AlF₄). **E** SEC elution profiles of hHsp90 after different time points of incubation with AMP-PNP at 37 °C (black: 0 min, blue: 1 h, red: 4 h,

green: 16 h). **F** Correlation of yHsp90 (left) and hHsp90 (right) ATPase activity with varying buffer salts and concentrations. Measurements were normalized to the activity of the respective protein in 200 mM KCl. Data is presented as mean values +/− SD from technical replicates (*n* = 3). **G** The obtained closing kinetics of yHsp90 (left) and hHsp90 (right) with 2 mM ATPγS for different buffer conditions. Kinetics were calculated by fitting SEC elution profiles at different time points of nucleotide incubation with a bi-Gaussian fit to determine the open- and closed-state fractions. Obtained kinetics are color coded according to **F**. (yHsp90: 200 mM KCl, $k_{closing}$ = 1.03 min⁻¹; 500 mM KCl, $k_{closing}$ = 1.94 min⁻¹; 500 mM (NH₄)₂SO₄, $k_{closing}$ = 1.26 min⁻¹; hHsp90: 200 mM KCl, $k_{closing}$ = 0.103 min⁻¹; 500 mM KCl, $k_{closing}$ = 0.177 min⁻¹; 500 mM (NH₄)₂SO₄, $k_{closing}$ = 0.439 min⁻¹). All measurements were performed as technical replicates (*n* = 3) to allow calculation of the mean and standard deviation.

differences with yHsp90 having a shorter N-terminal domain and a shorter linker compared to hHsp90. Both proteins seem to have a common mechanism for the conformational changes during the ATPase cycle[37–39]. However, in contrast to the yeast protein, open states have been predominantly detected for hHsp90[37,40]. Furthermore, the proteins differ regarding their optimum temperatures and ATP hydrolysis rates[41–43]. hHsp90 hydrolyzes approximately 0.1 ATP min$^{-1}$ whereas yHsp90 displays a tenfold higher activity of 1 ATP min$^{-1}$ [2,44,45]. In addition, the human Hsp90 possesses an expanded set of co-chaperones.

Here, we set out to investigate the inherent evolutionary differences of the two proteins and we further elucidate the human Hsp90 ATPase cycle. We show that yeast and human Hsp90 differ in their conformational dynamics and specific changes associated with the formation of the closed state. Furthermore, pronounced differences in the ATPase rate and closing reaction are pinpointed to two highly conserved residues in which yeast and human Hsp90 differ.

## Results

### Nucleotide dependence of the closing mechanism of Hsp90

The conformational changes of Hsp90 during its ATPase-driven reaction cycle have been extensively studied for yeast Hsp82[30]. The structural rearrangements induced by nucleotide binding lead to the closing of the dimer, which brings the NTDs into close proximity. Despite a sequence identity of over 60% and high structural similarity, human and yeast Hsp90 differ drastically in their ATPase activity and optimal temperature of operation (Fig. 1B). Furthermore, while the closed state is readily accessible for yHsp90, special conditions such as crosslinking or high salt concentrations have been required to induce a stable closed state for hHsp90[40,46]. To track nucleotide-induced conformational transitions of Hsp90 under physiological conditions (pH 7.5, 200 mM KCl), we used size-exclusion chromatography coupled to multi-angle light scattering (SEC-MALS). When we incubated yHsp90 (Hsp82) or hHsp90 (Hsp90β) with the slowly hydrolysable ATP analog ATPγS, both exhibited a significant shift in the elution volume while their molecular weights remained unchanged. Thus, their hydrodynamic volumes were decreased, indicating the formation of the closed state (Fig. 1C). A similar effect was also observed when the sedimentation behavior was monitored by analytical ultracentrifugation (Supplementary Fig. 1). Here the proteins displayed a shift toward higher Svedberg values after incubation with ATPγS. Since the sedimentation coefficient is largely affected by the displaced solvent of a given particle, this confirms the notion of a more compact state of Hsp90 in the presence of ATPγS. Finally, limited proteolysis experiments showed increased protection of Hsp90 in the presence of ATPγS against α-chymotrypsin as compared to the protein without added nucleotide (Supplementary Fig. 2). Taken together, the presence of ATPγS induces a compact, closed conformation in both yeast and human Hsp90 suggesting that the general mechanism of ATP-induced structural alterations is conserved.

To test the effect of different adenosine nucleotides on the closing reaction, SEC experiments were performed (Fig. 1D). Without added nucleotide or in the presence of ADP, the open state was the predominant species for both yHsp90 and hHsp90. Moreover, we observed no shift toward a more compact closed form upon incubation with ATP. This is in line with the prevailing notion that Hsp90 populates mainly open conformations, even during the ATPase cycle[47,48], and that the closed state is present during only a small fraction of the reaction time. Similar to ATPγS, the non-hydrolysable ATP analogue AMP-PNP lead to a significant shift in elution of Hsp90 due to the decreased hydrodynamic volume of Hsp90 upon closing (Fig. 1D). Therefore, both ATPγS and AMP-PNP trap Hsp90 in the closed state by slowing/preventing cycle progression, resulting in the accumulation of the closed conformation. Interestingly, yHsp90 and hHsp90 exhibit a drastic difference in their nucleotide specificity for

AMP-PNP. Whereas yHsp90 displayed a complete closing after 30 min, this reaction required overnight incubation for hHsp90 to achieve the same effect (Fig. 1E), suggesting that the process is less favorable for hHsp90. However, with ADP-AlF$_4$, which serves as a proxy for the trigonal bipyramidal hydrolysis transition state of ATP[49], the two Hsp90s did not show any changes in the elution profile. Therefore, the standard effects of ADP-AlF$_4$ would be consistent with the notion that the opening of the dimer occurs during the transition of the γ-phosphate from the tetrahedral conformation into the trigonal bipyramidal pre-hydrolysis state. However, currently, it cannot be ruled out that, due to the composite ATPase site of Hsp90, ADP-AlF$_4$ is not the ideal mimetic for the transition state and therefore may not unambiguously report on the role of the transition state during opening.

### Hydrophobic interactions promote the formation of the closed state

Dimerization of the N-terminal domains is essential for the ability of Hsp90 to hydrolyze ATP[15]. After nucleotide binding and closing of the ATP lid, the repositioning of the hydrophobic surfaces of the α1-helix and of the catalytic loop promotes the formation of the closed state[35]. By altering the ionic composition of the buffer, we aimed to modify the underlying interactions and consequently modulate the ATPase activity. We selected three salts from different regions of the Hofmeister series[50] that modulate hydrophobic interactions: ammonium sulfate on the kosmotropic (strengthening) side, calcium chloride on the chaotropic (weakening) side, and potassium chloride in the middle. Moreover, we tested different concentrations of the respective salts to differentiate the impact of the ionic compounds. Our data show that increasing the concentration of potassium chloride from 200 to 500 mM resulted only in a minor increase in ATPase activity for yHsp90, whereas hHsp90 showed a more significant gain of activity (Fig. 1F, Supplementary Fig. 3). Contrary to our expectations, the use of ammonium sulfate did not improve the ATPase activity of yHsp90 as compared to potassium chloride, although we observed a weak correlation between the ammonium sulfate concentration and the yHsp90 activity. For hHsp90, on the other hand, ammonium sulfate led to a pronounced increase in the ATPase activity as compared to potassium chloride. Between 200 mM and 500 mM ammonium sulfate, the ATPase increased from 10% up to 50% (with respect to 200 mM KCl). When calcium chloride was present in the buffer, both yHsp90 and hHsp90 showed a substantial decrease in activity (Supplementary Fig. 3). The chaotropic salt interferes with the hydration shell of the protein, highlighting the importance of the hydrophobic effect on closing. However, one should bear in mind that Ca$^{2+}$ could form a complex with ATP at high concentrations which may affect the results of the ATPase assay.

We then performed SEC experiments to compare the effects of the different salts on the Hsp90 closing kinetics. Fitting the elution profiles with a bi-Gaussian function allowed us to quantify the open and closed populations at each time point (Fig. 1G). Using this approach, a closing rate of $k_{closing} = 1.025$ min$^{-1}$ and $k_{closing} = 0.103$ min$^{-1}$ was calculated for yHsp90 and hHsp90, respectively (in 200 mM KCl). In agreement with the difference in ATPase activity, the hHsp90 closing kinetics are ten times slower than for yHsp90. Thus, in both cases, the conformational changes leading to the closed state seem to be rate-limiting. An increase in the potassium chloride concentration leads to a significant increase in the closed state population and the ATP catalysis rate (yHsp90 $k_{closing} = 1.941$ min$^{-1}$; hHsp90 $k_{closing} = 0.177$ min$^{-1}$). As potassium is on the kosmotropic side of the Hofmeister series, this demonstrates the importance of hydrophobic interactions during the closed state formation of the dimer.

We additionally tracked the conformational changes by chemical crosslinking experiments with disuccinimidyl glutarate (DSG). When the Hsp90 protomers are in close proximity, they have a higher

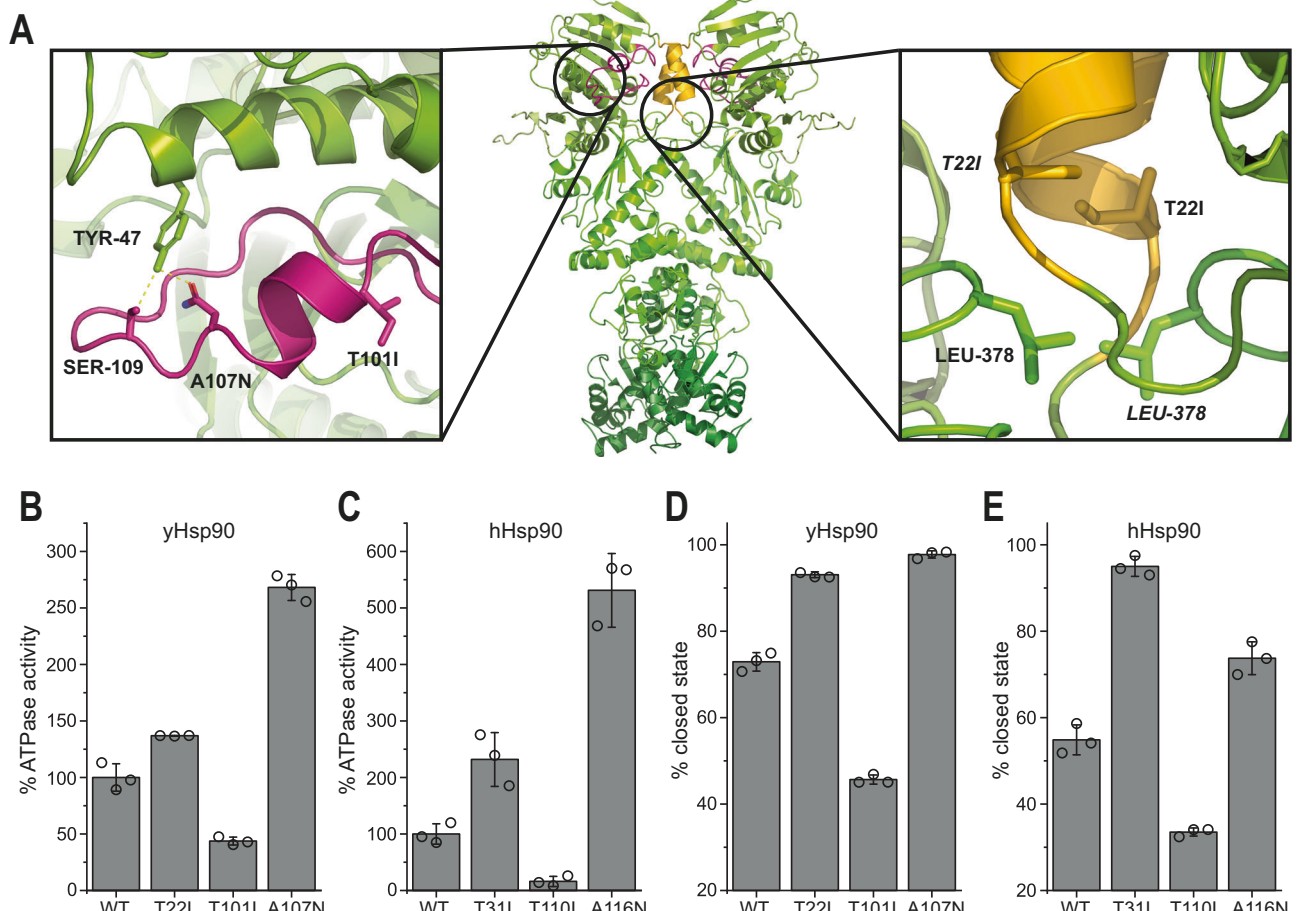

**Fig. 2 | Mutational analysis in the NTD reveals important interactions for Hsp90 ATPase activity and closing. A** Modified crystal structure of yHsp90 (PDB ID: 2CG9) highlighting mutations affecting ATPase activity. ATP lid (magenta) and α1-helix (yellow) are highlighted according to Fig. 1A. **B** Comparison of wt yHsp90 and mutant ATPase activities. The obtained activity was normalized to wt yHsp90. **C** Comparison of analog hHsp90 mutants. **D** wt yHsp90 closed-state fraction after 1 min of incubation at 30 °C with 2 mM ATPγS compared to mutants. Data were obtained by fitting SEC elution profiles with a bi-Gaussian fit. **E** Wt hHsp90 compared to analog mutants after 10 min of incubation at 37 °C. All measurements were performed as technical replicates ($n = 3$) to allow calculation of the mean and standard deviation.

possibility of being crosslinked by DSG as compared to the open state, and can be identified as a distinct band by SDS-PAGE (Supplementary Fig. 4). Our time-resolved crosslinking in the presence of ATPγS revealed similar kinetics of the closed state formation as obtained from SEC elution analysis (hHsp90 $k_{closing} = 0.148$ min$^{-1}$; yHsp90 $k_{closing} = 1.63$ min$^{-1}$). Furthermore, we find that yHsp90 exhibits a higher fraction of crosslinked dimer at $t = 0$ min, suggesting that the closed state is more readily accessible. In line with the results of the ATPase activity measurements, treatment with ammonium sulfate did not lead to a strong increase in the closing rate for yHsp90 ($k_{closing} = 1.26$ min$^{-1}$) and a larger fraction of the open population was present at any given time point. Even after longer incubation times with ammonium sulfate, we could not observe a complete closing of the yHsp90. However, hHsp90 displayed a considerable increase in the closing kinetics with ammonium sulfate ($k_{closing} = 0.432$ min$^{-1}$).

Taken together, the different effects salts have on yeast and human Hsp90 hint toward an evolutionary change between yHsp90 and hHsp90 concerning the transition from the open to the closed state.

### Effect of mutations in the NTD on the closing reaction
To test the notion that specific aspects of the closing reaction differ between yeast and man, we made use of several mutations, which had been shown to either positively or negatively affect the overall ATPase

activity of Hsp90 similar to the activation/inhibition by co-chaperones[51,52]. Most of these mutations are located in the NTD either near the ATP lid or near the α1-helix (Fig. 2A). In yHsp90, the T101I and A107N mutations (T110I and A116N in hHsp90) both affect lid movement. In this regard, T101I inhibits the ATPase activity by stabilizing the open state of the lid via hydrophobic interactions with the α1-helix[37,53], whereas A107N enhances ATPase activity by introducing a hydrogen bond between Asn-107 and Tyr-47, stabilizing the lid in the closed state (Fig. 2B, C)[35,47,54]. Similarly, T22I (T31I in hHsp90) increases the hydrophobicity of the α1-helix, improving its ability to interact with the helix of the opposing protomer, and thus enhancing the ATPase activity. Similar effects of these mutations confirm the common underlying mechanism of the ATPase cycle for yHsp90 and hHsp90[37], including the lid rearrangement and dimerization through hydrophobic interactions via the α1-helix (Fig. 2C).

We find that the yHsp90 lid mutation, T101I, showed nearly an identical ratio of the open and closed population in the *apo* state and in the presence of ATPγS, highlighting the importance of the lid rearrangement for the closing of Hsp90 (Fig. 2D). In contrast, for the A107N mutation, nearly a complete transition to the closed state was observed. Moreover, the human Hsp90 lid mutants T110I and A116N showed a similar behavior as the yeast mutants, confirming the common mechanism of lid rearrangement upon nucleotide binding (Fig. 2E). Since these mutations do not alter the nucleotide binding, the

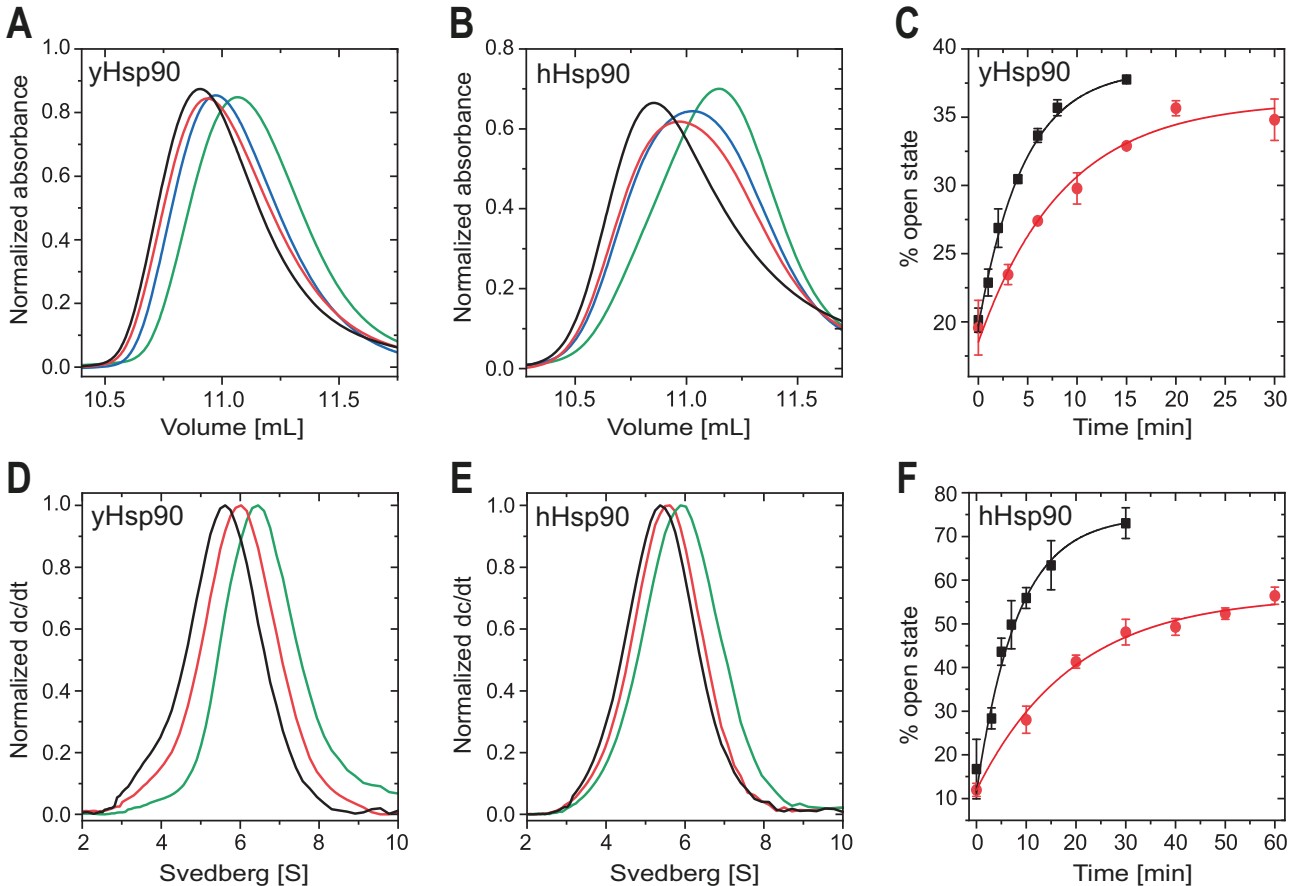

**Fig. 3 | Nucleotide exchange induced dimer opening. A** Size exclusion chromatography elution profiles of ATPγS closed yHsp90 (green) after additional incubation with 2 mM ATP (blue), ADP (red) or 500 μM radicicol (black) for 1 h at 30 °C. Shift to lower elution volumes indicate conformational rearrangements to a more open state. **B** Analog hHsp90 SEC elution profiles after incubation for 1 h at 37 °C. Peaks are normalized by peak area. **C** Obtained opening kinetics of yHsp90 and (**F**) hHsp90 with 500 μM radicicol (black) or 2 mM ADP (red). Kinetics were calculated by fitting SEC elution profiles with a bi-Gaussian fit to determine the open- and closed-state fractions (yHsp90 ADP $k_{opening}$ = 0.167 min$^{-1}$, radicicol $k_{opening}$ = 0.312 min$^{-1}$; hHsp90 ADP $k_{opening}$ = 0.0742 min$^{-1}$, radicicol $k_{opening}$ = 0.176 min$^{-1}$). All measurements were performed as technical replicates ($n$ = 3) to allow calculation of the mean and standard deviation. **D** Analytical ultracentrifugation sedimentation profiles of Atto488 labeled yHsp90 and (**E**) hHsp90 re-opened with ADP (red) or radicicol (black).

combined results indicate that lid rearrangement is not only necessary for the formation of the closed state, but also is a key factor for determining the closing kinetics. The increased hydrophobicity of the α1-helix mutant T22I resulted in a distinct increase in closing for yHsp90 and hHsp90. A direct comparison of the effect of the α1-helix mutant demonstrates a more severe impact on hHsp90. Similar to the influence of ammonium sulfate on the closing kinetics, this reveals a strong dependence of hHsp90 on hydrophobic interactions during the formation of the closed state. Although T31I exhibited the fastest closing kinetics for hHsp90, the ATPase of the A116N lid mutant remained significantly higher, suggesting that the lid and α1-helix mutants impact the closing mechanism of yeast and human Hsp90 differently.

We also probed the influence of co-chaperones on the closing kinetics in the presence of ATPγS (Supplementary Fig. 5): For both yHsp90 and hHsp90, we observed accelerated closing when incubated with Aha1 and Sti1/Hop substantially inhibited the formation of the closed state. Although Sba1/p23 is well-known as an ATPase inhibitor, the co-chaperone did not exhibit a significant effect on the closing reaction.

Overall, these findings demonstrate that both yHsp90 and hHsp90 depend on proper lid rearrangement and the formation of hydrophobic interactions for ATPase cycle progression. However, when hydrophobic interactions were enhanced through buffer composition or mutation, the closing kinetics of hHs90 were more strongly affected and a significant increase in ATPase activity was observed, highlighting an evolutionary difference between the two proteins.

## Nucleotide exchange induces dimer re-opening

The general mechanistic understanding of Hsp90 involves the assumption that ATP hydrolysis or release of ADP/P$_i$ triggers the dimer re-opening and cycle progression[55,56]. However, increasing evidence suggests that nucleotide release in the closed state is an alternative explanation for the re-opening[47,57]. To investigate the underlying mechanism of nucleotide exchange and dimer re-opening, we probed the ability of various nucleotides to exchange the bound ATPγS. In these experiments, yHsp90 or hHsp90 were first closed using ATPγS, followed by 1 h incubation with ATP, ADP, or the Hsp90 inhibitor radicicol, followed by analysis of the elution profiles obtained by SEC. Radicicol led to the most significant shift toward lower elution volumes and thus an increase in the open conformation of both yHsp90 and hHsp90 (Fig. 3A, B). This indicates that nucleotides can trigger re-opening once the dimer is in a closed state. For ADP and ATP, we also obtained shifts toward the open conformation but to a lower extent as compared to radicicol. Hsp90 proteins incubated with ADP showed a more pronounced population of the open state compared to ATP. Opening with ADP could be observed even in the presence of inorganic phosphate (Supplementary Fig. 6). Strikingly, when AMP-PNP was used for closing, neither ADP, ATP nor radicicol increased the open population after incubation (Supplementary Fig. 7). These experiments

suggest that the binding of AMP-PNP differs from ATPγS and that the exact geometry of the phosphates plays a crucial role during closing and opening.

To confirm the re-opening data, we performed analytical ultra-centrifugation experiments with Atto488-labeled yHsp90 and hHsp90 pre-incubated with ATPγS to initiate closing, as previously described. Similar to the results obtained by SEC, the radicicol-treated proteins displayed the most significant shift toward lower a sedimentation range (Fig. 3D, E), indicating a less compact conformation of the proteins induced by the re-opening. The treatment of closed dimers with ADP did not lead to a complete decrease in the closed population, as observed in the SEC analyses. Therefore, it can be assumed that a fraction of the protein remained in the closed state due to the presence of ATPγS. To obtain a more detailed picture of this reaction, re-opening kinetics with ADP and radicicol were analyzed. Interestingly, yHsp90 and hHsp90 displayed similar kinetics (yHsp90 $k_{\text{opening}}$ = 0.312 min$^{-1}$; hHsp90 $k_{\text{opening}}$ = 0.176 min$^{-1}$) when incubated with radicicol (Fig. 3C, F). Since it can be excluded that ATPγS is hydrolyzed during the given timeframe, the observed conformational changes must be induced by the exchange of ATPγS with radicicol. For both proteins, the nucleotide exchange with ADP reaches an equilibrium, but leaves a fraction of dimer in the closed state. Again, similar kinetics of the exchange were observed (yHsp90 $k_{\text{opening}}$ = 0.167 min$^{-1}$; hHsp90 $k_{\text{opening}}$ = 0.074 min$^{-1}$). We find the most plausible explanation for these results is the formation of an affinity- and diffusion-based equilibrium, as the same concentration of ATPγS and ADP is present in the buffer. The higher population of the open state observed with radicicol is due to its nanomolar affinity for Hsp90[58].

To further investigate the inhibitory effect of p23/Sba1 on Hsp90, the co-chaperone was added to Hsp90 after the closed state had formed. The analysis of the opening reaction of Hsp90 by SEC showed that the presence of p23/Sba1 inhibited the radicicol-induced opening of the dimer (Supplementary Fig. 8) consistent with previous results[38].

## smFRET measurements reveal differences in the population and dynamics of conformational states between yHsp90 and hHsp90

To obtain insights regarding the conformational landscape of both Hsp90s under different nucleotide conditions, we performed single molecule FRET (smFRET) experiments using multiparameter fluorescence detection with pulsed interleaved excitation (MFD-PIE)[59]. For yHsp90, we used the previously described FRET mutant D61C[30] labeled with the Atto532 (donor) and Atto643 (acceptor) fluorophores attached to different monomers in the dimer. For hHsp90, the protein contains many accessible cysteines in the CTD that are important for the function of the protein. Hence, we used Sortase-based ligation of hHsp90 fragments to fluorescently label a cysteine introduced at position 70 to the NTD fragment. Afterward, the labeled NTD was ligated together with the MD-CTD fragment. The same donor (Atto532) and acceptor (Atto643) fluorophores were used as for yHsp90. The structural integrity, stability, basal ATPase activity, as well as the ability to form a closed state of the ligated and labeled hHsp90 construct (hHsp90_LPTKG) was confirmed by in vitro characterization in comparison to WT hHsp90 (Supplementary Figs. 1, 9).

We first determined the FRET efficiency of labeled yHsp90 and hHsp90 in the absence of nucleotides (Fig. 4A, B). For yHsp90, there was one distinct population with a FRET efficiency close to zero while hHsp90 showed a low FRET peak along with a broad distribution of higher FRET populations. To investigate the underlying FRET distributions in more detail, we analyzed the fluorescence lifetime information available from the MFD-PIE data (Supplementary Fig. 10B, Supplementary Table 1). Here, yHsp90 exhibited a single, static FRET population on the timescale of a burst, whereas, for hHsp90, we could distinguish three states with FRET efficiencies of -0.05, 0.2, and a minor fraction of FRET efficiencies above 0.9. Fluctuations are

observable between the 0.2 and 0.9 FRET-efficiency states, which is responsible for the broad distribution of FRET values. We then measured both Hsp90s under the conditions where closing was observed by SEC (Fig. 1D, right panel) to assess the FRET efficiency of the closed conformation. For yHsp90, an increase in FRET efficiency from near zero to 0.08 was observed in the presence of AMP-PNP, which also appears static on the millisecond timescale (Fig. 4A, C). The AMP-PNP-induced FRET value change to 0.08 corresponds to a distance of 89.6 Å, which is in good agreement with the previously reported value of 87.6 Å for the closed conformation in yeast[31]. In the presence of ATPγS, the low FRET efficiency peak is very similar to the *apo* state (i.e., the open conformation; Supplementary Table 1, Fig. 4D). Interestingly, analysis of the fluorescence lifetime distribution suggests that the closed conformation is also present, potentially in a dynamic equilibrium (Supplementary Table 1). A very minor, dynamic population fluctuating to a state with FRET efficiencies >0.7 is also observed. Fluctuations to more compact states have been reported previously for yHsp90 using smFRET with different labeling positions[60]. In the case of hHsp90, measurements with AMP-PNP and ATPγS showed a FRET efficiency of -0.19 indicating closing of the dimer (Fig. 4B, E, F). This is most apparent in the fits to the fluorescence lifetime distribution (Supplementary Table 1). As there were no similar smFRET results on hHsp90 to compare to, we performed accessible volume (AV) calculations to estimate the expected FRET efficiency and distance values based on the reported crystal structures[61]. For yHsp90, an expected distance of 88.8 Å between fluorophores was determined for the closed state (PDB ID: 2CG9), which compares very well with the measured FRET distance of 89.6 Å in the presence of AMP-PNP. The AV calculation for hHsp90 (PDB ID: 5FWK) yielded an expected FRET distance of 89.9 Å, which is also in good agreement with the measured values of ~82–86 Å for the closed dimer in the presence of AMP-PNP and ATPγS. Furthermore, the fluorescence lifetime analysis for hHsp90 in the presence of AMP-PNP and ATPγS revealed a second, dynamic population, which fluctuates between the closed state and a compact state or states with high FRET efficiency (>0.8) (Fig. 4E, F, blue and red lines). This indicates that hHsp90 is more dynamic than its yeast counterpart and that the NTDs undock and rotate, which transiently brings the fluorophores into close proximity, and as also supported by our molecular simulations (see below). To get an estimate of the timescale of the dynamics, we performed a dynamic photon distribution analysis (PDA) for both yHsp90 and hHsp90 (Fig. 4H, I, Supplementary Fig. 12, 13, Supplementary Table 2)[62]. The PDA analysis for Hsp90s in the presence of ATPγS revealed transition rates on the order of milliseconds, with the dynamic behavior much more prevalent for human Hsp90 (Fig. 4H, I).

To test whether the closing of the Hsp90s observed with smFRET follow similar kinetics as seen by SEC, we measured the emergence of the closed FRET populations after 5, 30, and 60 min for yHsp90 in the presence of AMP-PNP (Supplementary Fig. 13A) and every 30 min over 4 h for hHsp90 in the presence ATPγS (Supplementary Fig. 14B). These conditions provide the highest contrast in the FRET histograms for the respective systems. For yHsp90, we see a shift in FRET efficiency from 0–0.08 when measured after 5 and 30 min with AMP-PNP, which is in agreement with the timescales of the closing kinetics measured by SEC (Fig. 1E, left panel). Similarly, in the case of hHsp90, we see a decrease in the fraction of proteins in the open state and an increase in the fraction of molecules with a higher FRET efficiency with ATPγS within 4 h. Considering that SEC only captures the average populations, the smFRET experiments show similar trends for the various nucleotides.

After establishing the smFRET distribution for the closed conformation, we determined the smFRET distributions in the presence of ADP and ATP (Supplementary Figs. 10, 11). For both Hsp90s, the open state is prominent with ADP (Supplementary Figs. 10C and 11C) whereas broad, dynamic distributions with higher FRET efficiencies are present with ATP (Supplementary Figs. 10D and 11D). Indeed, ATP

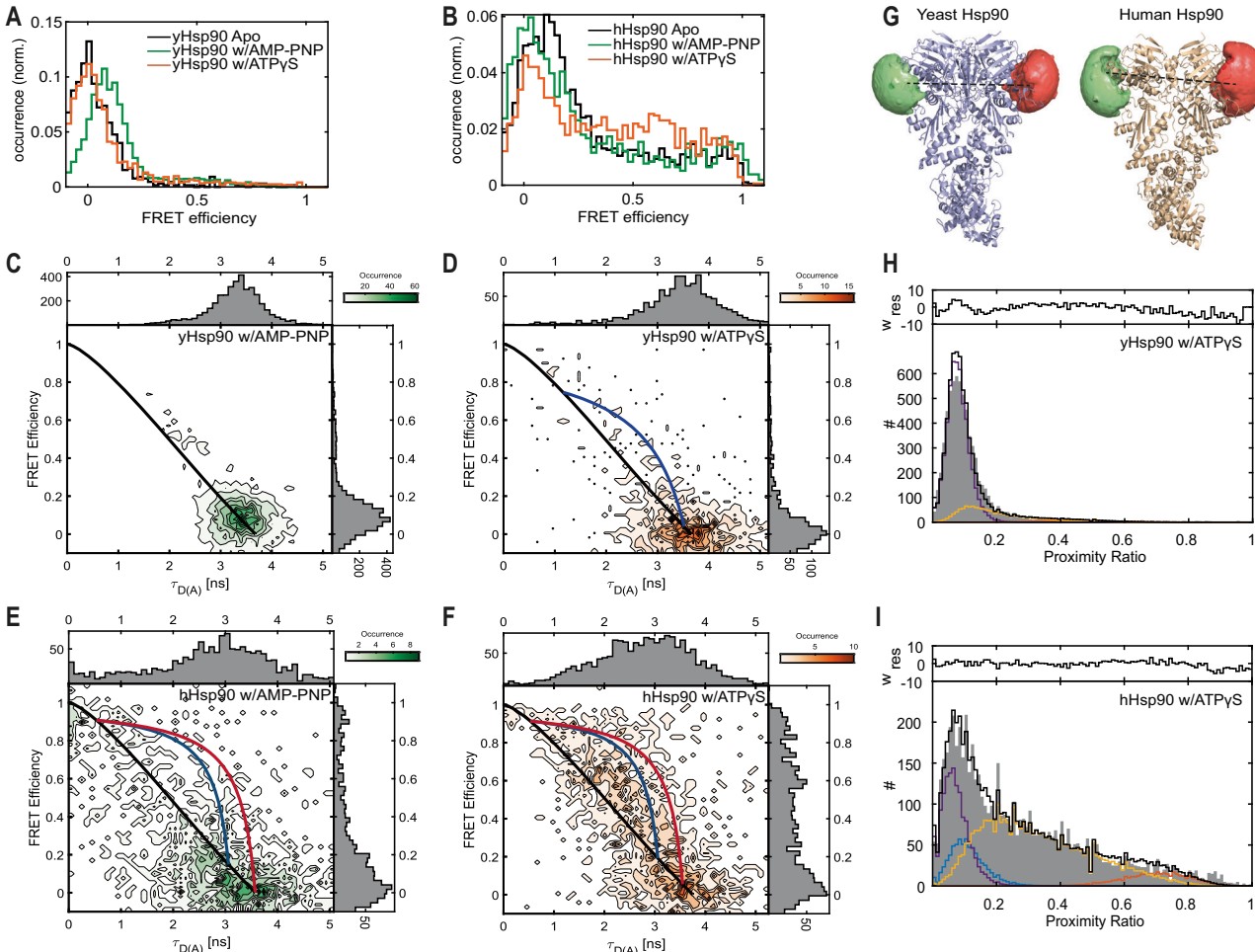

**Fig. 4 | Nucleotide dependent conformations and their dynamics for yHsp90 and hHsp90 measured by smFRET.** **A** SmFRET efficiency histograms of yHsp90 in the *apo* state (black), in the presence of AMP-PNP (green) and in the presence of ATPγS (orange). **B** SmFRET efficiency histograms of hHsp90 in the *apo* state (black), in the presence of AMP-PNP (green) and in the presence of ATPγS (orange). FRET histograms are of representatives from at least two independent measurements unless otherwise mentioned. **C**–**F** 2D histograms of FRET efficiency vs. donor fluorescence lifetime in the presence of an acceptor ($\tau_{D(A)}$) for **C** yHsp90 w/AMP-PNP, **D** yHsp90 w/ATPγS, **E** hHsp90 w/AMP-PNP and **F** hHsp90 w/ATPγS. Black lines indicate the static FRET line whereas both the red and blue curved lines depict dynamic FRET lines. **G** Accessible volume (AV) calculations of residues C61 for yHsp90 (left panel) and residue C70 for hHsp90 (right panel) labeled with Atto532 as a donor and Atto643 as an acceptor dye. Distances of 88.8 Å (AV) versus 89.6 Å (measured) for yHsp90 and 89.9 Å (AV) and 82–86 Å (measured) for hHsp90 were determined. **H**, **I** Dynamic photon distribution analysis (PDA) for yHsp90 and hHsp90, respectively, in the presence of ATPγS. The open conformation is shown in violet, the closed conformation in blue, a compact conformation with high FRET efficiency in red and the contribution of dynamically interconverting populations are shown in yellow. For experiments with yHsp90, nucleotides were preincubated with the protein for 2 h. In the case of hHsp90, the preincubation time was 4 h.

exhibited the highest conformational dynamics between the open and compact states for yHsp90 and between the closed and compact states for hHsp90 (Supplementary Figs. 10D and 11D, Supplementary Table 1). Besides ATP, all other nucleotide-bound states are static on the millisecond timescale for yHsp90. However, the presence of the closed state and dynamics are seen for all nucleotides tested with hHsp90. The observed dynamics are faster for hHsp90 than for yHsp90[63] (Supplementary Figs. 12, 13, Supplementary Table 2), suggesting that hHsp90 is much more flexible and dynamic than yHsp90.

## Conformational rearrangements involve all three domains

The exchange of hydrogen atoms with deuterium atoms (HDX) in the protein's backbone amide groups measures the solvent accessibility of amino acids and is sensitive to changes in protein conformation and stability[64,65]. By comparing the HDX patterns of Hsp90s in the open and closed state (Supplementary Figs. 15–19), we aimed to gain further insights into differences in the structural dynamics of the two proteins during the conformational rearrangements (Fig. 5A, B).

In the NTD of yHsp90 and hHsp90, we identified two main structural elements that display pronounced differences in HDX in the presence or absence of nucleotides: the α1-helix (yHsp90: residues 10-21; hHsp90: residues 19-30) and the ATP lid (yHsp90: residues 94-124; hHsp90: residues 103-132) (Fig. 5C, D). Interestingly, the α1-helix can be further divided into two regions with different exchange profiles. Residues oriented toward the β1-strand showed an increase in exchange, while residues near the catalytic loop showed a decreased exchange after closing. Upon nucleotide binding, the repositioning of the ATP lid over the binding pocket and the exposure of the N-terminal residues of the α1-helix[35] lead to an increase in exchange. Since the hydrophobic patch formed by the catalytic loop of the MD and the α1-helix is in contact with the other protomer in the closed state, hydrogen exchange in these structural elements is impeded. Strikingly, the lid displayed the highest decrease in exchange at the residues located near the hinges, highlighting the importance of these highly conserved residues. The conformational changes in the ATP lid and α1-helix are also independently supported by our molecular dynamic simulations.

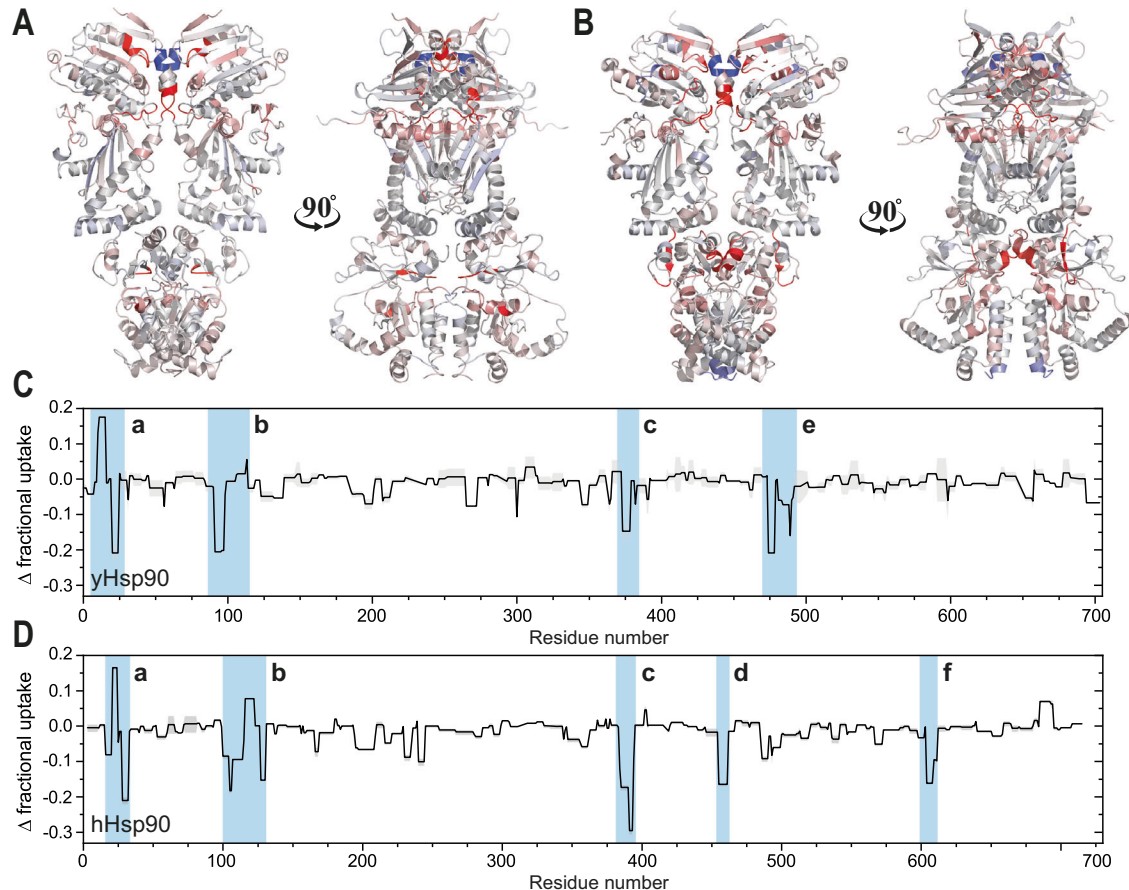

**Fig. 5 | HDX comparison of the open and closed state. A** Change (Δ) in fractional uptake between the open and closed state of yHsp90 plotted on its crystal structure in the closed state (PDB ID: 2CG9). Red indicates decreased exchange after closing, blue areas with increased exchange. **B** Change (Δ) in fractional uptake between the open and closed state of hHsp90 plotted on its crystal structure (PDB ID: 5FWK). **C** Fractional uptake (Δ) of yHsp90 and (**D**) hHsp90 plotted by residue.

Blue highlights crucial elements exceeding the significant threshold (a: α1-helix, b: ATP-lid, c: catalytic loop, d: M1/M2-hinge, e: MD/CTD-hinge, f: dimerization interface); gray areas indicate the standard deviation of the respective residues. All measurements were performed as independent technical replicates ($n = 2$) to allow calculation of the mean and standard deviation.

Both Hsp90 proteins also showed a significant decrease in HDX in the MD at residues around the catalytic loop upon closing (yHsp90: residues 373-383; hHsp90: residues 385-395). The arginine piston, which contacts the γ-phosphate of the nucleotide, is a key element for ATP hydrolysis and stabilizes the closed state[24]. Therefore, rearrangements repositioning the residues of the catalytic loop to form additional interactions with the NTD and the γ-phosphate of ATP result in reduced exchange. For hHsp90, the β-sheet connecting the subdomains of the MD, often referred to as M1 and M2, exhibited a notable decrease in exchange. Since yHsp90 did not show a similar behavior, during evolution, hHsp90 seems to have diverged by rearranging its MD during the ATPase cycle, which results in the different inherent enzymatic properties. In addition, a more pronounced decrease was visible for MD residues contacting the C-terminal domain (residues 494-498) in yHsp90 relative to hHsp90. As a consequence, both proteins differ significantly in the way the MD rearranges during closing. We also observed a slight decrease of HDX in helix 24 (residues 604-610) of the C-terminal domain of hHsp90 that forms an interface between the protomers. Hence, all three domains of the human protein undergo rearrangement during the ATPase cycle and hydrolysis.

In summary, the obtained HDX results confirm a common hydrolysis mechanism involving the α1-helix, the ATP lid, and the catalytic loop. However, variations in the HDX profiles, mainly in the MD, highlight differences regarding the structural rearrangements as a consequence of nucleotide binding.

## Differences in the dynamics of conformational changes between the human and yeast Hps90 revealed by molecular dynamics simulations

The experiments conducted emphasize the significance of the ATP lid and α1-helix in both the formation of the closed state and the overall ATPase activity of Hsp90. The essential role of these elements is highlighted by the nearly complete conservation of the corresponding residues between the yeast and human isoforms. Strikingly, the α1-helix and ATP-lid of yHsp90 and hHsp90 differ by one highly conserved amino acid each: Thr13 and Ser109 (Ala22 and Gln118 for human). Therefore, to probe the impact of the two given residues on the conformational dynamics of Hsp90, we performed atomistic molecular dynamics simulations by introducing the T13A/S109Q (mimicking A22 and Q122 in the hHsp90) mutations in silico in the full-length yHsp90 (PDB ID: 2CG9). We compared the molecular dynamics in both the ATP-bound closed conformation and in the *apo* state to probe possible molecular differences between the human and yeast Hsp90. In addition, we studied the dynamics of the NTD of hHps90 and yHsp90 in the ATP and *apo* states. Our simulations suggest that the global dynamics of the closed state of both Hsp90 isoforms are overall similar, but with distinct differences around the N- and M-terminal domains, particularly in the ATP-bound form, that could rationalize the experimental findings (Fig. 6A, Supplementary Figs. 14–16). In this regard, the data indicate that the *apo* form is generally more dynamic compared to the ATP bound form

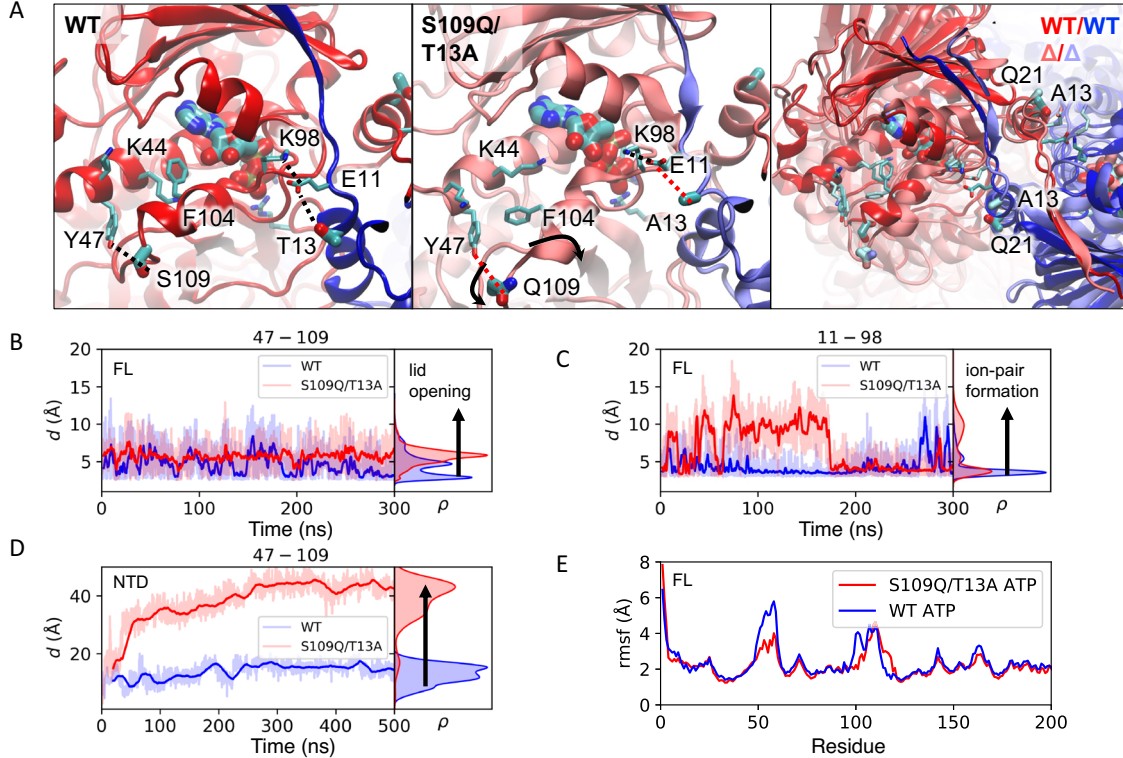

**Fig. 6 | Molecular dynamics (MD) simulations of WT and S109Q/T13A yHsp90.**
**A** MD simulations of the full-length yHsp90 suggest that the S109Q/T13A mutations result in conformational changes in the NTD, particularly around the ATP lid, and residues surrounding the ATP binding site. **B** Dynamic changes in the ATP lid (Tyr47-Ser/Gln109 distance) and **C** ion-pairs affecting Hsp90 dimerization (Glu11-Lys98 distance). **D** Introduction of the S109Q/T13A mutation results in an opening of the ATP binding site in simulations of the NTD construct (see Methods). **E** Conformational dynamics of the NTD (from full length yHsp90 simulations) indicated by the changes in root-mean-square-fluctuation (rmsf). See also Supplementary Figs. 14–16 for further analysis.

(Fig. 6, Supplementary Figs. 14–16). The ATP molecule forms tight interactions with the ATP lid (residues 94-125), Arg380 of the middle domain and the Glu33/Arg32 ion pair (Fig. 6A–C), which was previously suggested to modulate the catalytic activity of Hsp90[66]. The simulations further suggest that Ser109 forms a hydrogen bond that stabilizes the ATP lid, whereas the S109Q substitution results in a more dynamic lid when ATP is bound (Fig. 6A, B, D). Remarkably, in simulations of the NTD-yHsp90, the S109Q substitution leads to a complete opening of the ATP binding site (Fig. 6, Supplementary Figs. 19C, D, 20). In WT yHsp90, Tyr47 and Ser109 are stabilized by hydrogen-bonding interactions so that the helices turn toward each other when ATP is bound (Fig. 6A, B, D). As a consequence, Gln-109 and the loop around this residue flip outward in some of the trajectories, with the effect further propagating along the ATP-lid helix and affecting its dynamics. Our data reveal that the mutations could also lead to a somewhat larger distance between Lys44 - Phe104, and in addition, the two helices that establish the binding cavity for the adenine and the ribose opens up. In this regard, the hydroxyl group of Thr13 could be involved in positioning Glu11 that, in turn, may affect the Glu11/Lys98 ion-pair contact in the ATP lid (Fig. 6C). In contrast, with the Ala13-substitution, Glu11 moves toward the ATP lid, which in turn makes the Glu11/Lys98 ion-pair rather tight. Despite the overall high flexibility in the complete ensemble of some of these regions (Fig. 6C, Supplementary Figs. 15, 16), our simulations performed for the human and yeast NTD further support these perturbed contacts around the ATP binding site (Fig. 6E, Supplementary Fig. 21). We note that these shifted interactions could affect the ATP lid and the α1-helix of the NTD (13-22) via the α1-α1 contacts of the Hsp90 dimer, as also suggested by our HDX data, and indirectly supports the opening kinetics extracted from the SEC and crosslinking analyses. Interestingly, these shifted contacts seem to propagate to the N-terminal β-sheet of the NTD (1-9), which

makes contacts with the remaining β-sheets of the NTD. In addition, the shifted α1 of the NTD also effects the contacts around the 23-26 loop that interacts with the middle domain (residues 376-385).

We note that these local changes in the NTD arising from the amino acid substitutions also affect the dynamics of the Arg32/Glu33 ion pair (Supplementary Fig. 24). This, in turn, may tune the catalytic barrier for ATP hydrolysis and affect the global opening/closing dynamics[25,33,66–68], as also suggested by the ATPase activity (Fig. 2) and drastically increased the population of the open state (Figs. 3, 4).

Although our atomistic simulations convergence on the studied ns-µs timescales (Supplementary Fig. 23), they capture only the local structural and dynamic changes arising from the substitutions. To probe longer timescales, we supplemented our atomistic simulations with ca. 100 µs coarse-grained molecular dynamics (cgMD) simulations of the full-length yHsp90 from ref. 67. The cgMD allow us to capture a wide range of open conformations on an approximate level (see Method sections for details). The atomistic molecular dynamics simulations capture conformations with C61-C61 separations between 72.5 and ~75 Å. When incorporating the additional distance imposed by the linkers, this leads to FRET efficiencies that correspond to the measured FRET efficiency of ~8% (Fig. 4, Supplementary Fig. 9). In hHsp90, there is a slight decrease in FRET efficiency, consistent with the subtle variation in measured distances from the MD simulations arising when the amino acids S109Q/T13A are substituted in the NTD (Fig. 4, Supplementary Fig. 22). The smFRET experiments with hHsp90 also revealed the formation of highly compact closed states (Supplementary Fig. 12), producing FRET efficiencies of >50%. While the cgMD simulations are not accurate enough to determine exact populations of the different compact-closed NTD-NTD conformations, fluctuations are observed where the C61 positions of the dimer approach to within 40 Å. These states arise from the dissociation and rotation of the NTDs

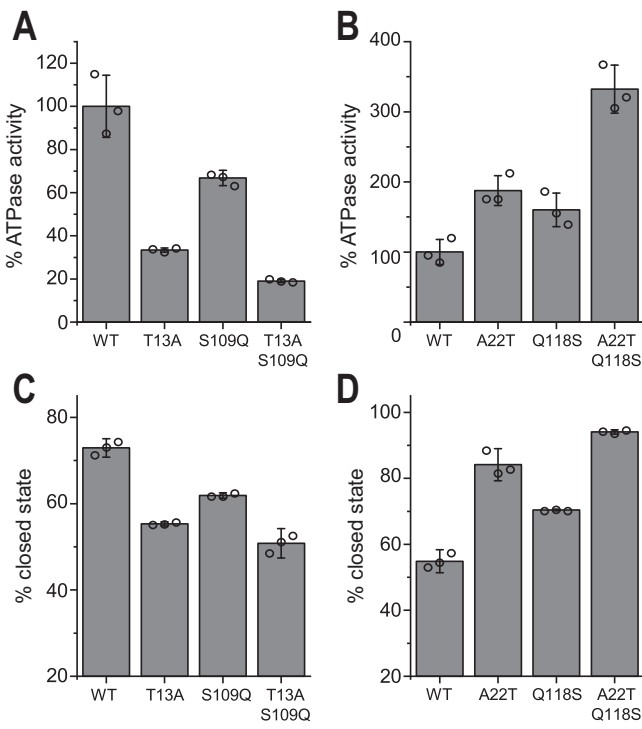

**Fig. 7 | Inherent evolutionary differences of yHsp90 and hHsp90. A** ATPase activity of wt yHsp90 and yeast-to-human mutants measured *via* a regenerative ATPase assay. **B** ATPase activity of wt hHsp90 and human-to-yeast mutants. **C** Calculated yHsp90 closed-state fraction after 1 min of incubation at 30 °C with 2 mM ATPγS compared to the mutants. Data were obtained by fitting SEC elution profiles with a bi-Gaussian fit. **D** Closed-state percentage of wt hHsp90 compared to mutants after 10 min of incubation at 37 °C. The ATPase activity was normalized to that of the respective wt protein. All measurements were performed as technical replicates (*n* = 3) to allow calculation of the mean and standard deviation.

(Supplementary Fig. 22), leading to higher FRET efficiencies (>50%). As the local substitutions in the NTD affect both NTD-NTD as well as NTD-MD interactions, these may shift the equilibrium population toward more of the compact-closed states, consistent with the experimental observations.

**Two point mutations are responsible for differences in closing kinetics and ATPase rate between yeast and human Hsp90**

To further test the effect of the T13A and S109Q mutations in vitro, the respective residues were exchanged in the yeast and human Hsp90s with their respective counterparts in the other species. The analysis of the mutants' ATPase activity revealed that yHsp90 T13A displayed a decrease in activity compared to the wildtype protein. In contrast, the A22T mutation in the human protein significantly increased its activity (Fig. 7). Similar observations were made when comparing the formation of the closed state. For yHsp90, the introduced mutation led to a slower formation of the closed state, whereas the exchange of this residue in human Hsp90, A22T, displayed an increase in the closing kinetics. For the second position, the yHsp90 variant S109Q led to a decrease of the ATPase rate by 32% compared to the wildtype protein. Additionally, a decrease in the closed population was evident for this mutant. For hHsp90, the Q118S mutation stimulated the ATPase activity by 57% and led to an increased closed state formation. However, when comparing the individual mutations, the mutation at position T13A/A22T had a more significant impact on the activity and N-terminal closing of the protein compared to S109Q/Q118S. Double mutants, containing both mutations, further enhanced the difference from the respective wildtype in the direction of the other Hsp90 species. Taken together and consistent with the molecular

dynamics simulations, the analysis of the chimera demonstrated that a major aspect of the difference in ATPase rate between yeast and human Hsp90 could be localized to two positions in critical structural elements. In particular, these results show that evolution specifically targeted the modulation of lid movement in Hsp90 to substantially change the function of the protein and adapt it to new needs.

## Discussion

Our combined analysis of human and yeast Hsp90 revealed important traits for understanding the evolution of the Hsp90 machinery. The picture emerging from these studies is that between yeast and human, Hsp90 has undergone evolutionary changes that influence structural transitions connected with the ATPase cycle. Notably, yHsp90 demonstrates a tenfold faster closing kinetics compared to the human protein. These findings confirm that the formation of the closed state represents the rate-limiting step of the ATPase cycle for both yeast and human Hsp90, in line with previous ensemble FRET measurements[30]. For hHsp90, the ATPase activity strongly correlates with the ion concentration and the effect of the respective salt on hydrophobic interactions according to the kosmotropic and chaotropic classification of the Hofmeister series[50]. In line with this notion, appropriate point mutations drastically influenced ATPase activity and closing kinetics[35,53]. The T31I mutation (T22I for yHsp90) increases the hydrophobicity of the α1-helix and the contacts with the hydrophobic residues of the catalytic loop. While this mutation leads to a 37% increase in activity for yHsp90, it causes a more substantial 141% increase in activity for hHsp90. In addition, T31I results in a higher percentage of the closed-state population for the human protein compared to yHsp90. These findings further support the notion that the closing mechanism of human Hsp90 evolved to be more strongly influenced by processes affecting the formation of hydrophobic interactions compared to yHsp90.

When AMP-PNP was used for closing, neither ADP, ATP, nor radicicol were able to trigger dimer re-opening for both yHsp90 and hHsp90. These findings suggest that the binding of AMP-PNP differs from ATPγS and that the γ-phosphate plays a critical role in the closing and opening process. Besides these common features, substantial differences in the formation of the closed state between yeast and human Hsp90 could be observed. Our molecular dynamic simulations in yeast suggest that small differences in the lid interactions can influence activity. Furthermore, the importance of the γ-phosphate was highlighted by the finding that ADP-AlF₄, which mimics the trigonal bipyramidal pre-hydrolysis state of the γ-phosphate[49], could not induce the closed state of Hsp90. The same observation was made when phosphate and ADP were present in the buffer. These results suggest that the structure of the γ-phosphate is an important element in the formation of the closed state and that dimer re-opening most likely occurs during the transition from the tetrahedral conformation to the trigonal-bipyramidal state of the phosphate. After this transition, the repositioning of the ATP lid eliminates the hydrophobic interactions between the α1-helices and the catalytic loop, leading to conformational changes that promote the formation of the open state.

The current understanding is that ATP hydrolysis occurring in the closed state initiates dimer re-opening and cycle progression[42,56]. Surprisingly, in our experiments, the addition of ADP, ATP or radicicol to closed Hsp90-ATPγS complexes resulted in the re-opening. This implies that the bound ATPγS can most likely be exchanged with ADP, ATP or radicicol. In this context, previous studies demonstrating the viability of the hydrolysis-deficient E33A Hsp90 mutant in yeast[47,57] support the notion that the exchange of bound ATP with ADP can serve as an alternative route for re-opening and cycle progression. However, the distinct mechanism behind the exchange remains elusive.

Our HDX analysis revealed that the open and closed states of yeast and human Hsp90 differ in the rearrangement of the MD and C-terminal domains. For yHsp90, closing includes changes in the

dynamics of several residues located in the hinge region between the MD and the C-terminal domain. In contrast, for human Hsp90, the results suggest that structural rearrangements of the MD subdomains are necessary for the closing process. In addition, the SEC elution volume of the closed population of human Hsp90 exhibited a more significant shift than for yHsp90, suggesting a greater disparity in the hydrodynamic volume between the two states. Therefore, the closed state seems to be more readily accessible for *apo* yHsp90 than human Hsp90, which is in line with previous studies[37,40]. This implies that hHsp90 may undergo more demanding conformational changes upon nucleotide binding and closing based on the increased conformational dynamics identified for human Hsp90 in our smFRET experiments. Whereas yHsp90 displayed deterministic structural re-arrangements upon nucleotide binding, the human protein exhibited a broad conformational distribution and increased flexibility. Conformational fluctuations to states with high FRET efficiency observed by the smFRET experiments for hHsp90 indicate that the NTDs undock and rotate, as also supported by our molecular simulations (Supplementary Fig. 22). These findings are in agreement with previous studies comparing the conformational dynamics of eukaryotic Hsp90s with HtpG from *E. coli*[38]. Thus, hHsp90 relies on a more flexible structure spending more time in the open state, which may allow the accommodation of a broader range of clients and provide an additional layer of regulation compared to yHsp90. Furthermore, the dynamic differences between hHsp90 and yHsp90 may support the regulation of hHsp90 by a more diverse set of co-chaperones[69].

The essential role of the ATP lid and the α1-helix is highlighted by the nearly complete conservation of the corresponding residues between the yeast and human isoforms. The α1-helix and ATP-lid of yeast and human Hsp90 differ by only one amino acid each: Ala22 and Gln118 (Thr13 and Ser109 for yeast). The created chimeric mutants with swapped residues altered the activity of both proteins, with the yeast protein experiencing a decreased ATPase activity while hHsp90 activity increased. Molecular dynamics simulations revealed that replacing Thr13 with Ala in yHsp90 caused Glu11 to move toward the ATP lid, resulting in a tighter ion pairing between Glu11 and Lys98. This ion pair hindered the lid re-arrangement upon nucleotide binding, thereby inhibiting the ATPase activity. A similar effect was observed for the Gln118/Ser109 mutation. This mutation directly affects the lid stability of the closed state by introducing an additional hydrogen bond to Tyr47 in hHsp90. As a result, the human protein exhibited an increase in its hydrolysis rate and closing kinetics. Comparing the effect of both mutations on ATP hydrolysis and the closing rates of Hsp90 revealed that the A22T/T13A mutant had a more significant impact on the protein and that both mutations affect Hsp90. Altering both residues of the chimeric mutants to the respective yeast/human amino acid further amplified the observed changes in activity and closing. This leads to the assumption that the mutation affects lid movement and stability, which, in turn, affects closing via independent mechanisms. We note that while both the HDX experiments and molecular dynamics simulations support conformational changes in the ATP lid and the α1-helix, the experimental HDX profiles reflect changes in the global opening/closing populations, while our molecular simulations capture local changes that take place on μs timescales.

We note that the activity of Hsp90 is modulated by the NTD-MD interaction, particularly by the Arg32/Glu33 and Arg380 contacts[25,33,66–68]. It has been suggested that this interaction tunes the catalytic barrier for ATP hydrolysis and affects the opening/closing dynamics[66]. Taken together, this suggests that local differences between the yeast and human Hsp90 strongly affect the dynamics of these functional elements, and could, in turn, result in the observed large-scale structural changes observed experimentally.

The data obtained suggest that the two mutations in the NTD of yHsp90 lead to similar structural behavior as hHsp90 (and vice versa),

which exhibits a less stable closed state and hindered lid movement. The identified residues and coupling sites have a decisive role in modulating the large-scale conformational changes by a complex interplay between local changes in the NTD and the global dynamics of the protein. This is in line with the notion that, in Hsp90, some post-translational modifications affect local structural switches, which can result in long distance conformational effects that influence the functional properties of the protein[2,70,71].

In conclusion, our study sheds light on the evolutionary differences between yeast and human Hsp90 and highlights the adaptive changes that have occurred in the human protein. hHsp90 has evolved a more flexible structure, spending more time in the open state. We speculate that this adaptation allows the chaperone to accommodate a wider range of clients as well as co-chaperones and provides additional regulatory mechanisms[29,72,73]. The modulation of lid movement in hHsp90 represents a targeted evolutionary change that significantly alters its function. In addition, our findings emphasize the dynamic differences between hHsp90 and yHsp90 and provide insights into molecular basis of the conformational cycle and chaperone activity of Hsp90 as well as the adaptations of hHsp90 to meet the specific demands of human cells.

## Methods

### SEC-MALS
SEC-MALS measurements were performed on a Shimadzu HPLC LC20A system equipped with an SPD-20A UV absorption detector unit and a HELEOS II MALS detector (Wyatt Technology). Proteins were separated at a flow rate of 0.5 mL/min using a Supradex200 Increase 10/300 GL column (GE Healthcare). BSA was used as a standard for the calibration of various buffers. Proteins were applied at a concentration between 0.2 and 0.4 mg/mL in a volume of 100 μL. Buffers (200–500 mM KCl/CaCl$_2$/(NH$_4$)$_2$SO$_4$, 40 mM HEPES pH 7.5, 5 mM MgCl$_2$, 6 mM β-mercaptoethanol) were cooled to 4 °C prior to measurements to minimalize conformational changes during the run. Data acquisition was performed with the Astra software (Wyatt Technology). Elution profiles were fitted with OriginPro 2021b using a custom bi-Gaussian fitting function.

### Crosslinking
For crosslinking reactions, disuccinimidyl glutarate (DSG) was dissolved in DMSO at a concentration of 50 mM and diluted to 10 mM with closing buffer (40 mM HEPES pH 7.5, 200 mM KCl, 5 mM MgCl$_2$ and 6 mM β-mercaptoethanol). Proteins were incubated at a concentration of 0.3 mg/ml with or without 2 mM ATPγS at 30 °C (yHsp90) or 37 °C (hHsp90). Crosslinking was achieved by the addition of DSG to a final concentration of 2 mM. The reaction took place in the dark at room temperature for 45 min. The reaction was stopped by the addition of 200 mM Tris for 15 min. Samples were mixed with 5x Lämmli buffer, denatured and loaded onto SERVA TG Prime 8% or SERVA TG Prime 4–12% gradient gels. Dimer and monomer fractions were calculated using ImageJ v1.53k; fitting was performed with OriginPro 2021b.

### HDX measurements
Hydrogen/deuterium exchange mass spectrometry experiments were performed on an automated ACQUITY UPLC M-class system equipped with a Leap PAL RTC, a HDX manager and a Synapt G2-S ESI-TOF mass spectrometer. 30 μM protein samples in either 500 mM ammonium sulfate buffer (hHsp90) or 500 mM KCl buffer (yHsp90) were pre-incubated with or without ATPγS to initiate closing prior to the measurement. Afterward, samples were diluted in a 1:20 ratio with PBS buffer (pH = 7.4) containing 99.9% deuterium oxide (D$_2$O). Samples were exposed to the buffer for 0, 10, 60, 600, 1800 or 7200 s. The hydrogen/deuterium exchange was stopped and proteins were denatured by the addition of a quenching solution (200 mM Na$_2$HPO$_4$,

200 mM KH$_2$PO$_4$, 4 M GdmCl, pH = 2.3) in a 1:1 ratio at 1 °C. 50 μL of the diluted sample were transferred for the on-column pepsin digest (Waters Enzymate BEH Pepsin Column 2.1 × 30 mm) at 20 °C. The peptides were trapped and subsequently separated on a Waters Acquity UPLC BEH C18 1.7 μm Vangard 2.1 × 5 mm trapping-column and a Waters Aquity UPLC BEH C18 1.7 μM 1 × 100 mm separation-column at 0 °C to minimize back-exchange. Eluted peptides were analyzed by the in line time-of-flight mass spectrometer. Peptide ions were separated by drift time within the mobility cell prior to fragmentation by MSE. The raw data were processed using the Waters Protein Global Server (PLGS) and the DynamX 3.0 software.

### Mutagenesis
Single amino acid changes in yHsp90 and hHsp90 were introduced using the Q5 site-directed mutagenesis kit (New England Biolabs) following the manufacturers protocol. For yHsp90 the following mutants were generated: L15R, T22I, T101I, A107N, T13A, S109Q, T13A/S109Q. For human Hsp90, the following mutants were generated: L24R, T31I, T110I, A116N, A22T, Q118S and A22T/Q118S. Successful mutations were confirmed by DNA sequencing (Eurofins).

### Protein purification
Human Hsp90 (Hsp90β) and mutants were expressed in *E. coli* BL21 cells with an N-terminal 6x His-SUMO fusion tag. Harvested cells were lysed by sonication in 50 mM Na$_2$HPO$_4$/NaH$_2$PO$_4$ pH 7.8, 300 mM NaCl, 10 mM Imidazole, 1 mM DTT, protease inhibitor HP (SERVA) and DNaseI (AppliChem). Lysates were cleared via centrifugation at 40,000 g for 1 h at 6 °C. Lysates were loaded onto a 5 mL His-Trap FF column (GE Healthcare), washed with 10 column volumes 97% 50 mM Na$_2$HPO$_4$/NaH$_2$PO$_4$ pH 7.8, 300 mM NaCl, 10 mM Imidazole, 1 mM DTT and 3% 50 mM Na$_2$HPO$_4$/NaH$_2$PO$_4$ pH 7.8, 300 mM NaCl, 500 mM Imidazole, 1 mM DTT. Elution was initiated with 100% 50 mM Na$_2$HPO$_4$/NaH$_2$PO$_4$ pH 7.8, 300 mM NaCl, 500 mM Imidazole, 1 mM DTT. His-tagged SUMO protease was added to pooled fractions and buffer was exchanged to 50 mM Na$_2$HPO$_4$/NaH$_2$PO$_4$ pH 7.8, 300 mM NaCl, 10 mM Imidazole, 1 mM DTT by dialysis at 4 °C overnight. The cleaved SUMO tag and the protease were removed by loading the pooled fractions again on a 5 mL His-Trap FF column and the flow through was collected. The flow through (~20 mL) was diluted with 40 mM HEPES, 20 mM KCl, 1 mM EDTA, 1 mM DTT to a final volume of 120 mL. The dilution was loaded onto a RESOURCE™ Q 6 mL anion exchange chromatography column (Cytiva) and a gradient to 60% 40 mM HEPES, 1 M KCl, 1 mM EDTA, 1 mM DTT was applied. Fractions containing the eluted protein were pooled and concentrated to a final volume of 2 mL using an Amicon Ultra-15 with 30 kDa cutoff. In a final step, the protein was applied on a HiLoad 16/600 Superdex 200 pg column (Cytiva) equilibrated with 40 mM HEPES, 150 mM KCl, 5 mM MgCl$_2$. After gel filtration chromatography, protein purity was confirmed by SDS-PAGE and Coomassie staining. Pure fractions were pooled, concentrated and frozen with liquid nitrogen for further use.

yHsp90 (Hsp82) and corresponding mutants were expressed in *E. coli* BL21 cells with an N-terminal 6x His tag. Purification was performed analog to human Hsp90 proteins with the except that, after the first Ni-NTA affinity chromatography, eluted protein was directly diluted to a final volume of 120 mL with buffer containing 40 mM HEPES, 20 mM KCl, 1 mM EDTA, 1 mM DTT.

### Fluorescent labeling of proteins
Before analytical ultracentrifugation, hHsp90 and yHsp90 were labeled with Atto-488 maleimide (ATTO-TEC) using a 1:2 protein/dye ratio. For yHsp90, a mutant carrying an introduced cysteine at position 61 was used. Human Hsp90 was expressed in two fragments (residues 1-265 & 272-724), which enabled site-specific labeling of an introduced cysteine (position 70). The labeled NTD, including the LPKTG motif, was ligated to the MD-CTD fragment using Sortase 5 M afterward.

Labeling was performed in a buffer containing 40 mM HEPES pH 7.5, 200 mM KCl, 5 mM MgCl$_2$ and 6 mM β-mercaptoethanol for 2 h at room temperature. The reaction was quenched by the addition of 2 mM DTT. Unreacted dye was separated from the labeled protein following the gravity protocol for PD-10 desalting columns (GE Healthcare) equilibrated at the same buffer without DTT. The labeled protein was concentrated using an Amicon Ultra-4 Centrifugal Filter Unit with 30 kDa cutoff (Millipore). The degree of labeling (~0.25) was determined by UV-VIS spectroscopy using the formula given on the manufacturer's homepage.

### Analytical ultracentrifugation
To analyze conformational changes of yHsp90 and hHsp90 upon closing and opening, analytical ultracentrifugation was performed by using a Beckman ProteomeLab XL-A centrifuge (Beckman) equipped with an AVIV fluorescence detection system. Atto488 labeled proteins at a concentration of 500 nM were either incubated with 2 mM ATPγS in 40 mM HEPES pH 7.5, 200 mM KCl, 5 mM MgCl$_2$ and 6 mM β-mercaptoethanol for 1 h at 37 °C or directly used after sample preparation. To initiate the opening of hHsp90, the protein was further incubated 1 h with either 2 mM ADP or 500 mM radicicol. For yHsp90, incubation times were reduced to 30 min and the temperature to 30 °C, respectively. Measurements were performed at 20 °C in a total volume of 300 mL and at 42.000 rpm in an An-50 Ti rotor (Beckman). The obtained data were imported into SedView to generate dc/dt profiles. Further analysis to acquire sedimentation coefficients and normalization was carried out in OriginPro 2021.

### Limited proteolysis
0.25 mg/mL hHsp90 was incubated with 2 mM ATPγS in buffer containing 40 mM HEPES pH 7.5, 200 mM KCl, 5 mM MgCl$_2$ and 6 mM β-mercaptoethanol for 1 h at 37 °C to initiate closing. For yHsp90, the incubation time was reduced to 30 min and the temperature to 30 °C. α-chymotrypsin (Sigma) was added to the protein at a 1:20 ratio (protease:protein). After various time points, samples were taken and the proteolysis reaction was quenched by the addition of 2 mM phenylmethylsulfonyl fluoride (Sigma). The digested samples were mixed with 5x SDS loading buffer, boiled at 95 °C and separated by SDS-PAGE on 4–12% acrylamide gradient gels (Serva) followed by Coomassie staining.

### ATPase activity assay
To measure ATPase activity in vitro, an ATP-regenerating system was used as described previously[74]. All measurements were carried out in closing buffer (40 mM HEPES pH 7.5, 200 mM KCl, 5 mM MgCl$_2$ and 6 mM β-mercaptoethanol) at 37 °C for hHsp90 and 30 °C for yHsp90, respectively. For the salt dependent ATPase activity assay, the buffer composition was changed replacing 200 mM KCl with the corresponding salt at its respective concentration. 60 μL of protein diluted in assay buffer (3 μM for yHsp90; 10–15 μM hHsp90) was mixed with 2x reaction premix containing 5.17 mM phosphenolpyruvate, 0.43 mM NADH, 6 units pyruvate kinase and 30.25 units of lactate dehydrogenase in the same buffer. The reaction was started by adding ATP to a final concentration of 2 mM. NADH depletion was monitored at 340 nm for at least 20 min in a Varian Cary 50 UV-VIS Photometer (Varian) or a Tecan Sunrise microplate reader (Tecan). The background activity of the sample was measured after the addition of 500 μM radicicol and subtracted from the original slope. Measurements were carried out in triplicates. Linear fits, means and standard deviation were calculated in OriginPro 2021.

### Atomistic molecular dynamics simulations
To probe the functional dynamics of the human and yeast Hsp90 isoforms, MD models were derived based on the crystal structure of the closed form of the yHsp90-Sba1 complex with ATP (PDB ID: 2CG9),

as well as for the NTD of the hHsp90α (PDB ID: 7L7J) and yHsp90 (PDB ID: 2CG9). For the full-length Hsp90 complex, the co-chaperone was removed, and missing parts of the amino acid sequence, including the charge-linker region, were modeled with MODELLER v. 10.2[75]. An $Mg^{2+}$ ion was modeled in the ATP binding site, with all ionizable residues modeled in their standard protonation states. Starting models for the S109Q/T13A substitutions were introduced in the mutation wizard of PyMOL[76]. The systems were embedded in a rectangular TIP3P water box, extending 15 Å in each direction, and with a 150 mM concentration of NaCl to neutralize the model. The full-length Hsp90 models (with ~300.000 atoms) as well as the NTD models (with ~75.000 atoms) were simulated in both their ATP-bound and *apo* states. All simulations were equilibrated stepwise, first for 50,000 steps (using a 2 fs integration timestep) with fixed protein, followed by 250,000 steps with flexible sidechains, and 1,000,000 steps without restraints. The MD simulations were performed using the CHARMM36m (c36 July 2020 update) force field[77,78] at $T = 310$ K using a 2 fs timestep and Langevin dynamics with a damping factor of 2 ps$^{-1}$, whereas the Nose-Hoover Langevin piston was used for pressure control ($p = 1.01325$ bar). Periodic boundary conditions were utilized to model long-range electrostatic interactions with the PME approach (grid size of 1 Å). All systems were simulated for 300–500 ns in three replicates (900–1500 ns in total) using NAMD2.14 or NAMD3[79] (see Supplementary Table 3 for list of MD simulations). The trajectories were analysed using Visual Molecular Dynamics (VMD) (1.9.4)[80], cpptraj (1.80)[81], MDAnalysis (2.4.1)[82] and mdtraj (1.9.7)[83].

## Coarse-grained molecular dynamics simulations

Coarse-grained molecular dynamics (cgMD) simulations from refs. 67,84. were analysed to compare with the smFRET experiments. Briefly, the cgMD simulations were created based on the atomistic model of yHsp90 using the MARTINI3 coarse-grained force field[85]. The model was embedded in a 300 Å cubic box with coarse-grained water and 100 mM NaCl, comprising ca. 165,000 beads. The simulations were performed in an NPT ensemble with a 10 fs timestep at $T = 310$ K using GROMACS[86] coupled to PLUMED2[87,88]. The protein–water interactions were increased by 6% to provide a more balanced protein-solvation effect and better reproduce the SAXS data[89,90]. An AlphaFold2-based elastic network model (AF-ENM) was introduced between residues with a high per-residue confidence score (pLDDT > 90), as described in ref. 84, where the strength and connectivity of the network is determined based on the expected positional error predicted by AlphaFold2[91]. To study large-scale conformational changes, simulations were performed using parallel-biased metadynamics[92] with 48 walkers[93] to enhance the conformational sampling. Distances and angles between the individual domains, as well as the radius of gyration, were used as collective variables. The trajectories were reweighted based on SAXS data with a Bayesian/maximum entropy approach[94]. See ref. 67. for further technical details. The cgMD simulations comprised in total 96 μs.

## SmFRET measurements and data analysis

Fluorescently labeled yHsp90 and hHsp90 monomers were pre-incubated at 37 °C and a concentration of 1 μM for double-labeled dimer formation in buffer containing 20 mM HEPES, 200 mM KCl, and 5 mM MgCl$_2$ and 6 mM β-mercaptoethanol at pH 7.5. The formed Hsp90 dimers, labeled with Atto532 and Atto643, were diluted to a final concentration of 100 pM in the same buffer for smFRET experiments. The measurements with different nucleotides were done in the presence of 2 mM ATP, ADP, ATPγS or AMP-PNP, added both during incubations (2 h incubation time for yHsp90 and 4 h incubation time for hHsp90) and during the smFRET measurements. A custom-built confocal setup equipped with multiparameter fluorescence detection (MFD) and pulsed interleaved excitation (PIE)[59] was used to perform the smFRET experiments as previously described[95]. The labeled

samples were excited with 532 nm and 640 nm laser lines with 70 and 25 μW of powers measured after the objective, respectively. MFD-PIE allows for the determination of FRET efficiency, stoichiometry, fluorescence lifetime and anisotropy for each single-molecule burst simultaneously. Accurate FRET efficiencies (E) were determined using the formula:

$$E = \frac{F_{GR} - \alpha F_{RR} - \beta F_{GG}}{F_{GR} - \alpha F_{RR} - \beta F_{GG} + \gamma F_{GG}} \quad (1)$$

where $F_{GG}$, $F_{GR}$ and $F_{RR}$ are the background-subtracted fluorescence signals detected in green/donor (GG), red/acceptor after donor excitation (GR) and acceptor channels (RR), respectively. For the Atto532-Atto 643 FRET pair, we used a Förster radius of 59 Å. Sub-millisecond dynamics of the Hsp90 dimers were visualized by analysing the FRET efficiency versus donor fluorescence lifetime in the presence of acceptor plots. For a static system, there is a linear relationship between the FRET efficiency (E) and fluorescence lifetime of the donor in the presence of the acceptor ($\tau_{D(A)}$) according to the formula:

$$E = 1 - \frac{\tau_{D(A)}}{\tau_{D(0)}} \quad (2)$$

Where $\tau_{D(0)}$ is the fluorescence lifetime of donor-only species. The assessment of dynamics between states were based on the E versus $\tau_{D(A)}$ plots where the theoretical static FRET line was defined according to the formula given in Eq. 2 with a slight modification of account for linker dynamics at high FRET efficiencies[62]. If the protein switches between two states within the duration of the burst, the calculated FRET efficiency will be a species weighted value dependent on the respective FRET efficiencies and the time spent in the respective states, while the donor fluorescence lifetime will be determined using a photon-weighted average. Therefore, FRET efficiencies (E) for each state are calculated according to the formula:

$$E = (\tau_1 * \tau_2 / \tau_{D(0)}[\tau_1 + \tau_2 - \tau]) \quad (3)$$

Where $\tau$ is the photon weighted average of the donor lifetime, and $\tau_1$ and $\tau_2$ are the fluorescence lifetimes of the two different states. The higher number of photons collected from the lower-FRET efficiency state leads to a deviation toward the right of the static FRET-line and is indicative of sub-millisecond dynamics. A lifetime analysis of the stoichiometry-filtered FRET populations was also performed to determine the number of subspecies and their respective FRET efficiencies (Table S1). A dynamic photon distribution analysis (PDA)[62] was done to further assess the FRET states and extract the timescales of the sub-ms transitions between different conformations (Table S2). PDA reveals the subpopulations and dynamics in a heterogeneous mixture of different states by investigating the photon statistics of the single molecule bursts that make up the width of the smFRET histograms. All the analyses of the collected data were done with the open-source PIE analysis with MATLAB (PAM) software[96].

## Accessible volume calculations

The accessible volume calculations for both Hsp90s shown in Fig. 4G were done with the FRET-restrained positioning and screening (FPS) software[61]. For yHsp90 (PDB ID: 2CG9), the C61 residues on each monomer and, for hHsp90 (PDB ID:5FWK), C70 residues on each monomer were used for donor-acceptor dye attachments. For the labeling dyes Atto532 and Atto643, the three radii AV model was used with a Förster radius of 59 Å, dye linker-length of 21 Å and linker width of 4.5 Å. The dye parameters used for AV simulations were $R_{532}(1)$: 5.5 Å, $R_{532}(2)$: 4.5 Å, $R_{532}(3)$: 1.5 Å for Atto532 and $R_{643}(1)$: 7.15 Å, $R_{643}(2)$: 4.5 Å, $R_{643}(3)$: 1.5 Å for Atto643.

**Reporting summary**

Further information on research design is available in the Nature Portfolio Reporting Summary linked to this article.

## Data availability

Data are available from the authors upon request. Source data are provided with this paper.

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

## Acknowledgements

This project was supported by SFB 1035 project A03 (to J.B.) and A11 (to D.L.), Cancerfonden (pj200968 to V.R.I.K.), the Knut and Allice Wallenberg Foundation (2019.0043 and 2019.0251 to V.R.I.K.). We are thankful for the computing time provided by SuperMuc at the Leibniz Rechenzentrum (project: pn98ha, pr83ro), and the National Academic Infrastructure for Supercomputing in Sweden (NAISS 2023/6-128) and Swedish National Infrastructure for Computing (SNIC 2022/1-29, LUMI project id: 465000179).

## Author contributions

S.R. and J.B. designed the study, S.R. performed cloning, protein production and analyses, M.R. did the yeast analysis and the AUC experiments, C.B. devised protein labeling strategies, G.A., E.B., and D.L. performed and analysed single molecule experiments, F.T. designed the initial Sortase constructs, A.M. purified some of the proteins, V.H., A.J. performed M.D. simulations, V.H., A.J., V.R.I.K. analysed the results, V.R.I.K. directed the computational work, S.R. and J.B. wrote the manuscript with discussions and contributions from all co-authors.

## Funding

## Competing interests

The authors declare no competing interests.
