## [Peer Review File · Nature Communications]

Evolution of the conformational dynamics of the molecular chaperone Hsp90REVIEWER COMMENTS

Reviewer #1 (Remarks to the Author):

In the manuscript “ Evolution of the conformational dynamics of the molecular chaperone Hsp90” Bucher and coworkers employed a battery of biophysical and computational approaches to show that the conformational transitions coupled to the ATPase cycle of Hsp90 are conserved from yeast to humans, but the dynamics is different. In contrast to yeast Hsp90, the human Hsp90 is characterized by broad ensembles of conformational states, irrespective of the absence or presence of ATP. The authors made a strong claim that the differences in the ATPase rate and conformational transitions between yeast and human Hsp90 are based on two residues in otherwise conserved structural elements that are involved in triggering structural changes.

Major Points

1) The submitted work presents an informative and well-executed study of the intersection between evolution and conformational dynamics of the Hsp90 chaperones. The biological problems addressed in this work are of clear fundamental and therapeutic interest and insights from computational approaches are certainly welcome to improve our understanding of the molecular mechanisms in this very important system.

2) The important focal point of this work is the realization that differences in the ATPase rate and conformational dynamics could be attributed to two highly conserved residues in which yeast and human Hsp90 differ. Despite a significant amount of data and analysis, the authors need to discuss this point in more detail, perhaps writing a separate subsection in which they consolidate the analysis of multiple experiments to validate this claim. The Hsp90 chaperones are highly dynamic and allosterically regulated machines where long-range couplings and allosteric interactions play a significant role in modulating the dynamics. How does this point of view play together with the proposed mechanism in which only two residues determine evolutionary-specific dynamic changes ?

3) The HDX experiments are important to this study but are discussed very briefly by leaving the reader somewhat puzzled about comparison of the HDX patterns of Hsp90s in the open and closed state. In particular, the authors suggested that the HDX results confirm a common hydrolysis mechanism involving the α 1-helix, the ATP lid, and the catalytic loop. At the same time, many differences in the HDX profiles were found, mainly in the MD. How do these differences affect the hydrolysis mechanism and what is the role of evolution in using these variations to induce species-specific conformational landscape ?

4) The central and perhaps most impressive and detailed part of the work is smFRET analysis that reveals differences in the population and dynamics of conformational states between yHsp90 and hHsp90. I think that the central results of smFRET studies should be spelled out more clearly. While the employment of MD simulations is certainly an important component of the work, it is relatively poorly integrated with HDX and especially smFRET experiments. It is not clear how MD data can be directly related to the dynamic insights from smFRET and how integration of HDX, smFRET and MD provides a more convincing rationale for the main conclusions.

5) The authors should better integrate MD results in the context of smFRET experiments. In the absence of a strong link between the results, the employment of MD tools may seem a bit artificial in the context of the entire work. I recommend the authors attempt to reorganize a bit the narrative and strengthen the connection between the experiments and computations.

6) How do the authors actually utilize MD simulations to “probe a possible molecular mechanism underlying the drastic kinetic and thermodynamic differences between the human and yeast Hsp90”. The trajectories reflect most local changes and at best may rationalize the thermodynamic changes but not kinetics of the process.

7) It was not obvious from this study whether the quality and length of MD simulations would have an impact on the results. Are computational predictions sensitive to fluctuations of MD trajectories, or perhaps more coarse-grained elastic network models would have been equally robust and perhaps more appropriate?

8) The manuscript needs to provide a statistical analysis to evaluate the significance and quality of MD predictions.

9) Could the authors clearly identify what makes their findings novel? What do the results of this study add to our current knowledge of the role of Hsp90 dynamics in catalytic mechanisms?

Reviewer #2 (Remarks to the Author):

The authors present an interesting body of work, which has been rigorously performed, delineating the physical features accounting for the difference in ATPase activity between the highly conserved yeast and human Hsp90 molecular chaperones. There is an ~10-fold difference in the ATPase rate between yeast and human Hsp90. The authors take a systematic approach to delineate why this is including classic salt-based biochemistry to modelling to deuterium exchange mass spectrometry. In the end, the authors narrow it to 2 residues that account for the majority of the rate differences and accompanying structural transitions. Importantly, they find that human Hsp90 rearranges its middle domain distinctly during the ATPase cycle resulting in different enzymatic properties. While the study is very well done and the results bring about interesting hypothesis on why this difference evolved, enthusiasm for the work is tempered a bit since the work does not actually identify an empirical reason for the ~10-fold decrease in the ATPase for the human Hsp90 compared to yeast. In considering the work for publication, the following points should be weighed:

1. In testing for the dimer re-opening the authors probed the influence of ATP, ADP, and radicicol on the exchange of ATP γ S and opening of the yeast and human Hsp90 dimers. While there is some interest in the impact of just ADP, the actual test should be with ADP+Pi. The authors should check the influence of ADP+Pi.

Furthermore, the exchange reaction should be validated using a radiolabelled ATP γ S to empirically confirm that the exchange has indeed occurred.

2. As the ligated human Hsp90 used in the smFRET experiments does not represent the full length protein (a fragment comprising amino acids 1-265 and 272-724), the basic activities of the ligated, truncated protein should be shown and compared to wild type human Hsp90 including basal ATPase activity and opening/closing behavior induced by salt and ATP as monitored by SEC-MALS.

3. Minor point: The authors observe a good correlation between ATPase activity and the closed state for wild type yeast and human Hsp90 (Figure 2). In addition, several mutants further support the conclusions reached except for the A116N mutant of human Hsp90. Might the molecular modelling used by the authors provide a possible explanation for why the A116N mutant doesn't follow the trend?

4. Minor point: The authors write "Surprisingly, the known ATPase inhibitor Sba1/p23 did not have a significant effect on the closing reaction" but is this "surprising" given the available structural and functional data on Hsp90 and p23? The use of "Surprisingly" should be reconsidered.

5. Minor point: There are grammatical errors/typos in the text (e.g., page 7 paragraph 2 "Our data shows.." should be "Our data show."; page 11 paragraph 2 "remains" should be "remained")

Reviewer #3 (Remarks to the Author):

The manuscript assesses the differences in the conformational cycle of yeast and human Hsp90 and identifies two key residues that are responsible for the major difference in the catalytic ATPase rate of these Hsp90 proteins. You conclude that the human protein is in a more open state than the human protein and that hHsp90 is more amenable to client protein interaction, enabling interaction with additional clients and co-chaperones.

Major concerns.

1. There is no evidence presented that the human protein is engaged with more client proteins and co-chaperones. Thus, the statement in the abstract that the human protein is more amenable to client protein interaction, enabling interaction with additional clients and co-chaperones should be rephrased. There could be other reasons that the human protein spends more time in a 'more open' state.
2. As far as I can tell you do not specify explicitly which Hsp90 isoform you are using. Is the data relating to Hsp90 alpha or beta protein?
3. On page 7 of the manuscript, you mention that when CaCl₂ is present in the buffer, both yHsp90 and hHsp90 show substantial decreases in activity. However, you do not refer to any figures that I can see for this result. Please refer the reader to the data.
4. NH₄SO₄ had a very different effect on the ATPase activity of yHsp90 and hHsp90. This was attributed to increased closing rates. I would also like to see SEC experiments showing that NH₄SO₄ shifts the hHsp90 to the closed state but not the yHsp90. This would eliminate any other unknown effects the NH₄SO₄ may have on the assay.
5. On page 8 of the manuscript, you claim that the A107N and T101I mutants of yeast Hsp90 affect the ATPase activity positively or negatively. I think a reference is required here to direct readers to that work. On page 8, you do not acknowledge prior work on Aha1 activation, HOP/Sti1 inhibition or p23/Sba1 inhibition.
6. Ala 31 is actually T31 in human Hsp90 beta. This is a highly conserved residue position in Hsp90 and I do not understand why you say the residue is Ala 31.

7. Similarly yeast Ser109 is Glutamine, not glutamate (Gln 118) in hHsp90 beta. Unfortunately, I could not assess the accuracy of your data without knowing the correct mutations. One other point, is there any evidence that Ser109 is phosphorylated? Could the faster rate of closer be due to that lack of phosphorylation at this position in the yeast protein and also the same argument could be had for T13 in yeast. What effect would phosphorylation at these sites have on the yeast protein?

8. Between residues 7 and 46 of yeast Hsp90, there are also a few other differences in sequence relative to hHsp90 beta. Specifically at positions 7 and 38. How do these affect ATPase/conformational dynamics? Why were these not considered in the analysis? It is, therefore, a bit premature to state that only these two mutations are involved in differences in the ATPase activity between yeast and human Hsp90. This needs to be further investigated or at least discussed.

Minor points

1. In Figure 2 A. Rather than give ASN-107 and Ile-101, the mutations should be labelled as A107N and T101I, so it is clear which are the mutations in the figure.

Reviewer #4 (Remarks to the Author):

The ATP-dependent 90 kDa heat shock proteins (Hsp90s) are essential molecular chaperones that assist folding and maturation of client proteins involved in regulating cell homeostasis, proliferation, differentiation, and cell death. The Hsp90 dimer promotes client folding and maturation via an ATP-hydrolysis-driven conformational cycle. In this cycle, the dimer transitions from an “open” V-shaped conformation, in which the N-terminal nucleotide-binding domains are far apart, to a “closed” conformation where the two N-terminal domains dock onto each other enabling ATP hydrolysis. Yeast is frequently used as model organism to study general cell biological processes believed to be conserved through eukaryotic evolution and relevant for human diseases as well. This is also true for the molecular chaperone Hsp90, for which the yeast protein is investigated in much greater detail than the human homologs and it is believed that the mechanism is conserved between yeast and human Hsp90. However, the yeast and human proteins present a 10-fold difference in ATPase rate for which the molecular cause and consequences are unclear.

Riedl and co-workers investigated in depth the differences between yeast and human Hsp90, analysing conformational changes in the two proteins using size-exclusion chromatography (SEC), analytical ultracentrifugation, single molecule Förster Resonance Energy Transfer (FRET), hydrogen-

exchange mass spectrometry and molecular dynamics simulations with wildtype and mutant proteins. The authors studied the opening and closing dynamics of human and yeast Hsp90 in the presence of different nucleotides and different salts of the Hofmeister series. They show that differences in ATPase rate between the two proteins depend on similar but distinct closing mechanisms of the N-terminal lid and the α 1-helix. The authors further investigated the influence of known mutations affecting Hsp90's ATPase activity and closing dynamics. The results presented in the manuscript demonstrate that human Hsp90 is more sensitive to changes in hydrophobicity in both the environment and the protein itself. Recently, the notion that ATP hydrolysis is essential for Hsp90 activity was challenged by reports demonstrating that nucleotide exchange is sufficient for maintaining activity in hydrolysis defective Hsp90. The manuscript explores this new concept by investigating ATP γ S displacement by ATP, ADP or radicicol using SEC and analytical ultracentrifugation as readout for the conformation. An elegant Sortase-based ligation method allowed the authors to fluorescently label the cysteine-rich hHsp90 allowing them to compare the closing dynamics of yeast and human protein also in single-molecule FRET measurements. These measurements revealed different kinetics and closing dynamics between the two proteins. Furthermore, hydrogen exchange experiments on the closed and open Hsp90 revealed that the differences in closing dynamics for human and yeast Hsp90 do not lie only in the N-terminal domain but also involve the middle and C-terminal domains. Finally, molecular dynamic simulations performed on the full-length yeast protein and both yeast and human N-terminal domains elucidated details of the N-terminal rearrangement. Interestingly, the authors were able to reverse some of the differences between the human and yeast proteins in the MD simulations by replacing only two residues (yeast T13A and S109Q) that differentiate the two proteins in the N-terminal domain. The same replacements introduced into yHsp90 reduced the ATPase activity almost to the level of wild type hHsp90. The introduction of the reverse mutations into hHsp90 increased the ATPase activity by 3.5-fold.

In summary, this work provides insights into the evolutionary adaptation of Hsp90 chaperones. It significantly expands our knowledge on the dynamics of the nucleotide-driven conformational changes in Hsp90 and how the ATPase cycle was tuned by single amino acid replacements during evolution. The results presented are exhaustive and state of the art and the conclusions largely justified. Given the widespread use of yeast as model system for protein folding diseases in humans, this study will meet considerable interest in the chaperone community. There are, however, some points the authors should address before publication.

Major concerns:

1. The manuscript presents many inaccuracies in the formatting and presentation of data. The quality of the scientific work displayed can be undermined by poor presentation. For examples see point 9 and points 5 and 8 in the "minor comments" section. We advise the authors to check the manuscript for potential mistakes that could have escaped this reviewer.
2. As one salt of the Hofmeister series the authors used CaCl₂ in concentrations up to 500 mM in their ATPase assays. Calcium phosphates are notorious for being poorly soluble (solubility limit for Ca₃(PO₄)₂, 64 μ M; for CaHPO₄, 316 μ M). How soluble is Ca-ATP? At 200 to 500 mM Ca²⁺ ions and 5 mM ATP, Ca-ATP could be precipitating, which could explain the reduced ATPase activity of Hsp90 as the residual ATP concentration in solution were considerably lower and Hsp90 has a rather high

KM for ATP (yHsp90: 350 μ M at 37°C, Wegele et al. 2003; hHsp90: 840 μ M at 37°C, McLaughlin et al. 2002). The authors need to address this question.

3. For the crosslinking experiments presented in Figure S4 it is not clear what the times shown in the figure refer to. If the time corresponds to the incubation in the presence of ATP γ S, how did the authors prevent further ATP γ S binding to Hsp90 during the 45 minutes DSG crosslinking? If the ATP γ S binding kinetics was quenched in one way or another (e.g. by lowering the temperature) the authors should show that this regime prevents binding and dissociation of ATP γ S. The authors should also clarify whether the crosslinking reaction was quenched with 10 mM Tris as stated in the figure legend or 200 mM Tris as stated in the methods section.

4. In Figure 2 B-C-D-E it is not clear how many replicates have been performed for each mutant and the ATPase activity of yeast mutant T22I seems to lack any error bars.

5. At the bottom of page 11, the authors refer to “ADP-induced opening”, however, the experiment they refer to in Figure S7 was performed using radicicol to induce Hsp90 opening, and not ADP, making the author's statement inaccurate.

6. For the smFRET measurement, the length of the incubation with nucleotides is never stated.

7. In their smFRET measurement in Figure 4A, the authors found for yHsp90 in the presence of AMPPNP a FRET efficiency of 0.08. Assuming a Förster radius of 60 Å (value missing in the manuscript; should be provided), these data are consistent with an average inter-dye distance of 88 Å as found in the crystal structure of AMPPNP bound yHsp90 with docked N-terminal domains. In contrast, for hHsp90 FRET efficiencies up to 1 were measured in the apo state and in the presence of ATP γ S or AMPPNP. Since in the presence of AMPPNP the N-terminal domains dock onto each other, it is difficult to understand how hHsp90 can exhibit FRET efficiencies up to 1, meaning an inter-dye distance of less than 40 Å. The N-terminal domains would need to rotate around the length axis of Hsp90 to bring the dye molecules closer together. This should be discussed in more detail.

8. For the HDX experiments, the authors should provide as supplementary information a list of all peptides analysed, their residue numbers and relative exchange under the different conditions used.

9. In the paragraph describing the MD simulations, there is some confusion of residue numbers and identity. For example, on top of page 18, the authors refer to hHsp90 Ala13 and Glu118 and the respective Thr22 and Ser109 in yeast while two lines further down the authors claim to have mutated T13A/S109Q in yeast mimicking A22 and Q122 of the human protein. Q stands for glutamine not glutamate (Gln not Glu). I hope the molecular modelling experts changed the correct residues.

10. From the observation that addition of ADP, Al³⁺ and F⁻ ions to Hsp90 does not lead to the closed conformation that authors conclude that “the opening of the dimer occurs during the transition of the γ -phosphate from the tetrahedral conformation into the trigonal bipyramidal pre-hydrolysis state”. Unless there is a crystal structure that shows that ADP·AlF₄ bound to yeast or human Hsp90 mimics the trigonal bipyramidal pre-hydrolysis state of ATP, this conclusion is not justified. The ATP binding pocket of Hsp90 is a composite binding site formed between NTD and

MD, two domains that are tethered to each other only by a flexible linker, with the NTD only binding the adenine, ribose and α and β phosphate and the MD contacting the γ phosphate, it is not clear whether ADP and AlF₄ form the ATP pre-hydrolysis mimetic. Just adding ADP and Al³⁺ and F⁻ ions to Hsp90 is no guarantee that the trigonal bipyramidal pre-hydrolysis state of ATP is trapped. The mitochondrial Hsp90 TRAP1 was crystallized with ADP-AlF₄ in the docked state strongly arguing against the claim of the authors (Lavery et al. 2014).

Minor comments:

1. The authors state “Heat shock proteins (Hsp) are a highly expressed, ubiquitous class of molecular chaperones”. This is an incorrect statement. Not all heat shock proteins are molecular chaperones, nor are all chaperones heat shock proteins!
2. The authors state “from yeast to human, Hsp90 has undergone evolutionary changes”. Such a statement implies that human Hsp90 descended from yeast, which is, of course, not the case. Both yeast and human Hsp90 descended from a common ancestor some 700 million years ago and evolved in different directions selected by different constraints. It may very well be that human Hsp90 serves a larger diversity of clients than yeast Hsp90 and therefore requires a higher flexibility. The authors should state this more correctly.
3. The author should define the error bars (standard deviation or standard error of the mean) in all figure legends of the manuscript and include in the panels an indication which protein, yeast or human Hsp90, was being analysed.
4. The authors should clearly state whether they are analysing human Hsp90 α or Hsp90 β .
5. The authors state that human and yeast Hsp90 have an 80% sequence similarity. Sequence similarity is not a rigorously defined term and the degree of sequence identity should be used instead.
6. In Figure 1A the authors should label the different structural elements to avoid confusion of the reader. In particular the catalytic loop is easily overlooked.
7. In supplementary figure S1 the X axes should refer to Svedberg rather than volume.
8. In supplementary figure S2 the authors should clarify whether time 0 represents a sample containing protease that is immediately quenched or a protease-free sample.
9. For Figure 1G did the author add any ATPyS or other nucleotides to obtain closure in the Hsp90s? This should be indicated in the figure legend.
10. Figure S3 misses the Y axes titles.
11. Figure 1 F and S3: For hHsp90 the ATPase activity at 500 mM KCl is higher than the ATPase activity at 200 mM in Fig. 1F but lower in Fig. S3. The x-axis labels seemed to be inverted in Fig. S3. Number of data points and significance should be indicated in all bar graphs.
12. In Figure 3 the labelling of the panels is likely wrong with C and E being inverted.

13. In supplementary figure S15 it is not clear why structures of full-length Hsp90 are provided next to the panel. What is their relation to understanding the dynamics of the N-terminal domains described in the figure?

14. In the description of the protein purification protocol, the authors state that a dilution to a final volume of 120 mL is performed but they never state the starting volume. Thus, the dilution factor is unclear. More clarity would help the reproducibility of the protein purification method described.

15. The authors abbreviate the closing rate with “kcat”. This is not logical as “cat” does not relate to “closing” and may lead to confusion as “kcat” is generally used as catalytic constant in steady state Michaelis-Menten type kinetics.

16. Panels D and F of Figure 6 are never referred to in the text.

17. The reference for Hofmeister does not seem to be correct.

Point to point reply to the reviewers' comments – Riedl et al.

Reviewer #1

In the manuscript “Evolution of the conformational dynamics of the molecular chaperone Hsp90” Bucher and coworkers employed a battery of biophysical and computational approaches to show that the conformational transitions coupled to the ATPase cycle of Hsp90 are conserved from yeast to humans, but the dynamics is different. In contrast to yeast Hsp90, the human Hsp90 is characterized by broad ensembles of conformational states, irrespective of the absence or presence of ATP. The authors made a strong claim that the differences in the ATPase rate and conformational transitions between yeast and human Hsp90 are based on two residues in otherwise conserved structural elements that are involved in triggering structural changes.

Major Points

1) The submitted work presents an informative and well-executed study of the intersection between evolution and conformational dynamics of the Hsp90 chaperones. The biological problems addressed in this work are of clear fundamental and therapeutic interest and insights from computational approaches are certainly welcome to improve our understanding of the molecular mechanisms in this very important system.

We thank the reviewer for the positive comments on the manuscript.

2) The important focal point of this work is the realization that differences in the ATPase rate and conformational dynamics could be attributed to two highly conserved residues in which yeast and human Hsp90 differ. Despite a significant amount of data and analysis, the authors need to discuss this point in more detail, perhaps writing a separate subsection in which they consolidate the analysis of multiple experiments to validate this claim. The Hsp90 chaperones are highly dynamic and allosterically regulated machines where long-range couplings and allosteric interactions play a significant role in modulating the dynamics. How does this point of view play together with the proposed mechanism in which only two residues determine evolutionary-specific dynamic changes ?

We agree with the reviewer that readers may benefit from a more integrated discussion of the evidence obtained from different experimental approaches on effects of the two mutations on Hsp90 ATPase and dynamics as well as their connection to long range allosteric coupling in Hsp90.

In the Discussion section, we have now added a combined presentation of the evidence. Furthermore, we have addressed the issue of local amino acid changes in the NTD and large-scale conformational effects on Hsp90. In the context of PTMs of Hsp90, it has been seen before that a change in a single amino acid side chain can act as a conformational switch that affects the entire protein. These aspects are included in this new section.

See also our answer to point 3.

3) The HDX experiments are important to this study but are discussed very briefly by leaving the reader somewhat puzzled about comparison of the HDX patterns of Hsp90s in the open and closed state. In particular, the authors suggested that the HDX results confirm a common hydrolysis mechanism involving the α 1-helix, the ATP lid, and the catalytic loop. At the same time, many differences in the HDX profiles were found, mainly in the MD. How do these

differences affect the hydrolysis mechanism and what is the role of evolution in using these variations to induce species-specific conformational landscape ?

The reviewer raises interesting points in the context of the HDX results. One conclusion we draw is that similar structural elements are affected in both yeast and human Hsp90 upon closing. The differences in other parts such as the MD are harder to explain based on our current understanding of Hsp90.

In this regard, we have clarified in the revised manuscript that the activity of Hsp90 is modulated by the NTD-MD interaction, particularly the Arg32/Glu33 and Arg380 contacts that tune the catalytic barrier for ATP hydrolysis and affect the opening/closing dynamics⁶⁷. The local changes in the NTD arising from the differences between the yeast and human Hsp90, also affect the dynamics of these functional elements, and may thus result in the global structural changes in Hsp90. We have added a discussion on these functional elements to the Discussion section of the revised manuscript.

Revisions in the main text:

“We note that these local changes in the NTD arising from the amino acid substitutions, also affect the dynamics of the Arg32/Glu33 ion pair (Figure S24), which in turn may tune the catalytic barrier for ATP hydrolysis and affect the global opening/closing dynamics^{25,33,67-69}, as also suggested by the ATPase activity (Figure 2) and drastically increased opening populations (Figures 3, 4).”

Addition to the Discussion section:

“We note that the activity of Hsp90 is modulated by the NTD-MD interaction, particularly by the Arg32/Glu33 and Arg380 contacts^{25,33,67-69} that have been suggested to tune the catalytic barrier for ATP hydrolysis and affect the opening/closing dynamics⁶⁷. Our combined data suggest that local differences between the yeast and human Hsp90 strongly affect the dynamics of these functional elements, and could in turn result in the observed large-scale structural changes observed experimentally.”

Additions in the SI:

“Figure S24. Dynamics of catalytically important elements in Hsp90.

A, B: Dynamics of the R32-E33 ion-pair in A) WT-NTD with ATP and B) S109Q-T13A-NTD with ATP. The S109Q-T13A substitution affects the dynamics of the R32-E33 ion-pair, leading to an increased population of the "open" ion-pair conformation.

C: Dependence of barrier for ATP hydrolysis on the R32-E33 distance from density functional theory (DFT) calculations (B3LYP-D3/def2-TZVP level). Data based on Ref. 4.

D: Snapshots on closed and open ion pair conformations from molecular dynamics simulations.”

4) The central and perhaps most impressive and detailed part of the work is smFRET analysis that reveals differences in the population and dynamics of conformational states between yHsp90 and hHsp90. I think that the central results of smFRET studies should be spelled out more clearly. While the employment of MD simulations is certainly an important component of the work, it is relatively poorly integrated with HDX and especially smFRET experiments. It is not clear how MD data can be directly related to the dynamic insights from smFRET and how integration of HDX, smFRET and MD provides a more convincing rationale for the main conclusions.

We thank the reviewer for recognizing the importance of these results and indicating that they should be emphasized more. We now integrate a discussion of the smFRET observations within the presentation of the MD results. Indeed, the course-grain simulations support the smFRET measurements demonstrating that the NTDs dissociate and rotate on the $\mu\text{s}/\text{ms}$ timescale. We have clarified that our atomistic molecular dynamics simulations bring valuable insight into how the substitutions affect the local conformations of key elements in the NTD, and their interactions with the surrounding subunits. However, they do not capture the large-scale conformational changes on the micro/millisecond-second timescales.

Based on the Reviewer's suggestions, we computed C61-C61 distance profiles and compared these to the results from the smFRET experiments. In order to explore longer timescales, we supplemented our atomistic simulations with longer (100 μs) coarse-grained molecular dynamics (cgMD) simulations, which indeed provide additional insight into the rich conformational dynamics of Hsp90.

Our combined analysis suggests that our atomistic simulations capture conformations that are consistent with states with a 8% FRET efficiency. We also mention that the subtle variations arising from the amino acid substitutions are consistent with the low FRET data, but these simulations do not account for the high FRET states with >50% efficiency. By extending the simulation time using cgMD simulations, we observe fluctuation consistent with the high FRET states suggesting that these states could arise from dissociation and rotation of the NTD. This has also been described before (Jussupow 2022, Daturpalli 2017, Lopez 2021). These findings suggest that the local substitutions in the NTD affect the NTD-NTD and NTD-MD interactions, as indirectly also supported by the HDX data.

Jussupow A, Lopez A, Baumgart M, Mader SL, Sattler M, Kaila VRI. Extended conformational states dominate the Hsp90 chaperone dynamics. *J Biol Chem.* 2022 Jul;298(7):102101. doi: 10.1016/j.jbc.2022.102101.

Daturpalli S, Knieß RA, Lee CT, Mayer MP. Large Rotation of the N-terminal Domain of Hsp90 Is Important for Interaction with Some but Not All Client Proteins. *J Mol Biol.* 2017 May 5;429(9):1406-1423. doi: 10.1016/j.jmb.2017.03.025.

A. Lopez, V. Dahiya, F. Delhommel, L. Freiburger, R. Stehle, S. Asami, *et al.* Client binding shifts the populations of dynamic Hsp90 conformations through an allosteric network. *Sci. Adv.*, 7 (2021), Article eabl7295

Addition in the main text:

"Although our atomistic simulations convergence on the studied ns- μs timescales (Figure S23), they capture only the local structural and dynamic changes arising from the substitutions. To probe longer timescales, we supplemented our atomistic simulations with ca. 100 μs coarse-grained molecular dynamics (cgMD) simulations of the full length yHsp90 from ref 67. The cgMD allow us to capture a wide range of open conformations on an approximate level (see

Method sections for details). The atomistic molecular dynamics simulations capture conformations with C61-C61 separations between 72.5 and ~75 Å. When incorporating the additional distance imposed by the linkers, this leads to FRET efficiencies that correspond to the measured FRET efficiency of ~ 8% (Figure 4, Figure S9). In hHsp90, there is a slight decrease in FRET efficiency, consistent with the subtle variation in measured distances from the MD simulations arising when the amino acids S109Q/T13A are substituted in the NTD (Figure 4, Figure S22). . The smFRET experiments with hHsp90 also revealed the formation of highly compact closed states (Figure S12), producing FRET efficiencies of >50%. While the cgMD simulations are not accurate enough to determine exact populations of the different compact-closed NTD-NTD conformations, fluctuations are observed where the C61 positions of the dimer approach to within 40 Å. These states arise from the dissociation and rotation of the NTDs (Figure S22), leading to higher FRET efficiencies (>50%). As the local substitutions in the NTD affect both NTD-NTD as well as NTD-MD interactions, these may shift the equilibrium population towards more of the compact-closed states, consistent with the experimental observations."

Additions in the SI:

Figure S22: Comparison of molecular simulations with smFRET experiments.

A: The D61-D61 distances, used as a proxy for the measured FRET efficiencies, taken from the atomistic molecular dynamics simulations suggest that the WT and T13A/S109Q of the FL-yHsp90 remain in an overall similar conformation to the closed-state X-ray structure (PDB ID: 2CG9). The distances are comparable to a state with ca. 8% FRET efficiency (see panel C), when including the additional contribution of the linkers and fluorophores. The T13A/S109Q mutation introduces a subtle local increase in the EFRET relative to the WT-yHsp90.

B: Coarse-grained molecular dynamics (cgMD) simulations on ~100 μs timescales show local dissociation and rotation of the NTDs leading to states with a short D61-D61 distance (< 65 Å), comparable with a high efficiency FRET (>40%).

C: General dependence of the FRET efficiency as a function of donor-acceptor separation with $R_0=59 \text{ \AA}$, based on $\text{EFRET} = 1/[1+(r/R_0)^6]$.

D: Snapshots from the cgMD simulations showing the canonical closed form of Hsp90 (left, comparable to $\text{EFRET} \sim 0.2$), and a structure with rotated NTDs (right, comparable to $\text{EFRET} \sim 0.9$). See Refs. 1-3 for further discussion on the NTD dissociation and rotation.

Addition to the Methods section:

Coarse-grained molecular dynamics simulations

Coarse-grained molecular dynamics (cgMD) simulations from Ref. ^{1,2} were analysed to compare with the smFRET experiments. Briefly, the cgMD simulations were created based on the atomistic model of γ Hsp90 using the MARTINI3 coarse-grained force field ³. The model was embedded in a 300 \AA cubic box with coarse-grained water and 100 mM NaCl, comprising ca. 165,000 beads. The simulations were performed in an NPT ensemble with a 10 fs timestep at $T = 310 \text{ K}$ using GROMACS ⁴ coupled to PLUMED2 ^{5,6}. The protein-water interactions were increased by 6% to provide a more balanced protein-solvation effect and better reproduce the SAXS data ^{7,8}. An AlphaFold2-based elastic network model (AF-ENM) was introduced between residues with a high per-residue confidence score ($p\text{LDDT} > 90$), as described in Ref. ¹, where the strength and connectivity of the network is determined based on the expected positional error predicted by AlphaFold2⁹. To study large-scale conformational changes, simulations were performed using parallel-biased metadynamics ¹⁰ with 48 walkers ¹¹ to enhance the conformational sampling. Distances and angles between the individual domains, as well as the radius of gyration, were used as collective variables. The trajectories were reweighted based on SAXS data with a Bayesian/maximum entropy approach ¹². See Ref. ² for further technical details. The cgMD simulations comprised in total 96 μs .

We note that both the HDX experiments and molecular dynamics simulations support that the substitutions lead to conformational changes in, e.g., the $\alpha 1$ -helix and the ATP-lid. However, the experimental HDX profiles reflect changes in the global opening/closing populations, which is difficult to model quantitatively based on our molecular simulations due to the different timescales. We have clarified in the revised text that despite the qualitative similarities observed in the HDX and molecular dynamics simulations in these elements, the quantitative characterization of these changes is challenging and is a topic of future work.

Addition to the Discussion section:

"We note that while both the HDX experiments and molecular dynamics simulations support conformational changes in the ATP lid and the $\alpha 1$ -helix, the experimental HDX profiles reflect changes in the global opening/closing populations, while our molecular simulations capture local changes that take place on μs timescales."

5) The authors should better integrate MD results in the context of smFRET experiments. In the absence of a strong link between the results, the employment of MD tools may seem a bit artificial in the context of the entire work. I recommend the authors attempt to reorganize a bit the narrative and strengthen the connection between the experiments and computations.

We thank the reviewer for this suggestion. To better compare the smFRET experiments with results from MD simulations, we included coarse-grained MD simulations over ca. 100 μs (from Jussupow et al, J Biol Chem 2022). The simulated and measured FRET histograms underscore the low FRET values of the open and closed states as well as transient

fluctuations of the NTD to higher FRET values. The information has been added to the MD results section and SI (Figure S22).

6) How do the authors actually utilize MD simulations to “probe a possible molecular mechanism underlying the drastic kinetic and thermodynamic differences between the human and yeast Hsp90”. The trajectories reflect most local changes and at best may rationalize the thermodynamic changes but not kinetics of the process.

As discussed above, we agree with the reviewer that the atomistic molecular dynamics simulations explore the local changes in the NTD and MD on microsecond timescales. We have revised the sentence to clarify the insight obtained from the MD simulations.

Revisions in the main text:

"Although our atomistic simulations convergence on the studied ns- μ s timescales (Figure S23), they capture only the local structural and dynamic changes arising from the substitutions."

"We note that while both the HDX experiments and molecular dynamics simulations support conformational changes in the ATP lid and the α 1-helix, the experimental HDX profiles reflect changes in the global opening/closing populations, while our molecular simulations capture local changes that take place on μ s timescales."

7) It was not obvious from this study whether the quality and length of MD simulations would have an impact on the results. Are computational predictions sensitive to fluctuations of MD trajectories, or perhaps more coarse-grained elastic network models would have been equally robust and perhaps more appropriate?

Per the reviewer's suggestion, we have now included coarse-grained simulations in the manuscript. The atomistic simulations provide valuable insight into the molecular details of the substitutions whereas the coarse-grained data provide additional insight into the large-scale conformational changes. We have performed these analyses and incorporated them in the revised text as discussed above.

8) The manuscript needs to provide a statistical analysis to evaluate the significance and quality of MD predictions.

We have added statistical analysis of the simulations, showing the RMSD convergence of the individual simulations (Figure S23), and that dynamics of the key distance are similar in the triplicate runs (Figure S19). The dynamics explored in the atomistic simulations converge locally, while the global dynamics of Hsp90 is better reflected within the cgMD simulations.

Changes in the SI:

Figure S23: Root-mean-square-deviations (RMSD) during molecular dynamics simulations.

The figure shows that the RMSD of the protein backbone locally stabilizes during the first few hundred nanoseconds of the simulation. A, B: RMSD of the NTD of WT-yHsp90 in (A) ATP and (B) apo states. C, D: RMSD of the NTD of T13A/S109Q-yHsp90 in (C) ATP and (D) apo states. E, F: RMSD of the NTD of WT-hHsp90 α in (E) ATP and (F) apo states. G, H: RMSD of the FL WT-yHsp90 in (G) ATP and (H) apo states. Extension of the simulations from 300 ns to 500 ns leads to small fluctuations in the overall structure, supporting that the (local) dynamics has converged. I, J: RMSD of the FL T13A/S109Q-yHsp90 in (I) ATP and (J) apo states.

9) Could the authors clearly identify what makes their findings novel? What do the results of this study add to our current knowledge of the role of Hsp90 dynamics in catalytic mechanisms?

Our study highlights the delicate balance between the local interactions in the catalytic site and their coupling to large-scale conformational changes in the global dynamics of Hsp90. The reported differences between the yeast and human proteins provide a basis for understanding the conformational cycle and chaperone activity of Hsp90.

We addressed the reviewer's query in the revised manuscript in the revised discussion as described above.

Reviewer #2

The authors present an interesting body of work, which has been rigorously performed, delineating the physical features accounting for the difference in ATPase activity between the highly conserved yeast and human Hsp90 molecular chaperones. There is an ~10-fold difference in the ATPase rate between yeast and human Hsp90. The authors take a systematic approach to delineate why this is including classic salt-based biochemistry to modelling to deuterium exchange mass spectrometry. In the end, the authors narrow it to 2 residues that account for the majority of the rate differences and accompanying structural transitions. Importantly, they find that human Hsp90 rearranges its middle domain distinctly during the ATPase cycle resulting in different enzymatic properties. While the study is very well done and the results bring about interesting hypothesis on why this difference evolved, enthusiasm for the work is tempered a bit since the work does not actually identify an empirical reason for the ~10-fold decrease in the ATPase for the human Hsp90 compared to yeast. In considering the work for publication, the following points should be weighed:

We appreciate the reviewer's comments and have addressed the queries below.

1. In testing for the dimer re-opening the authors probed the influence of ATP, ADP, and radicicol on the exchange of ATP γ S and opening of the yeast and human Hsp90 dimers. While there is some interest in the impact of just ADP, the actual test should be with ADP+Pi. The authors should check the influence of ADP+Pi. Furthermore, the exchange reaction should be validated using a radiolabelled ATP γ S to empirically confirm that the exchange has indeed occurred.

We conducted the experiment with ADP + Pi as recommended and incorporated the data into the revised version.

Our findings reveal that inorganic phosphate does not hinder Hsp90 from transitioning to the closed state with ATP γ S, nor does it impede the subsequent reopening with ADP. We have included details of this experiment in the supplementary information and updated the main text accordingly.

We are currently unable to perform exchange tracking with radiolabeled ATP γ S. However, our closing data together with the additional evidence strongly suggest that exchange occurs.

Additions to the main text:

“Opening with ADP could be observed even in the presence of inorganic phosphate (Fig. S6).”

2. As the ligated human Hsp90 used in the smFRET experiments does not represent the full length protein (a fragment comprising amino acids 1-265 and 272-724), the basic activities of the ligated, truncated protein should be shown and compared to wild type human Hsp90 including basal ATPase activity and opening/closing behavior induced by salt and ATP as monitored by SEC-MALS.

The reviewer's point shows that we may not have explained the ligation and the fragments used clearly enough. In the ligated construct, the sequence between positions 266 – 271 has been altered from KKKTKK in the wildtype to LPKTGA in the mutant. Consequently, the ligated protein retains the full-length Hsp90 structure, with only 4 residues changed at the ligation site.

A comprehensive analysis of the mutant's properties in comparison to the wildtype Hsp90 is shown in Figure S9 in the Supplementary Information of the revised manuscript. These experiments assess the structural integrity, thermal stability, basal ATPase activity, and closing behavior of the mutant Hsp90 compared to wildtype. Together, the evidence demonstrates that the mutation does not affect critical properties of the protein.

3. Minor point: The authors observe a good correlation between ATPase activity and the closed state for wild type yeast and human Hsp90 (Figure 2). In addition, several mutants further support the conclusions reached except for the A116N mutant of human Hsp90. Might the molecular modelling used by the authors provide a possible explanation for why the A116N mutant doesn't follow the trend?

Unfortunately, the detailed quantitative exploration of this substitution in silico and biochemically is beyond the scope of the present work, but will be addressed in future studies.

4. Minor point: The authors write "Surprisingly, the known ATPase inhibitor Sba1/p23 did not have a significant effect on the closing reaction" but is this "surprising" given the available structural and functional data on Hsp90 and p23? The use of "Surprisingly" should be reconsidered.

We thank the reviewer for this comment. We have changed the sentence accordingly avoiding the use of "surprisingly".

Revisions in the main text:

"Although Sba1/p23 is well-known as an ATPase inhibitor, the co-chaperone did not exhibit a significant effect on the closing reaction."

5. Minor point: There are grammatical errors/typos in the text (e.g., page 7 paragraph 2 "Our data shows.." should be "Our data show.."; page 11 paragraph 2 "remains" should be "remained")

Many thanks for bringing these to our attention. We have corrected the typos.

Reviewer #3

The manuscript assesses the differences in the conformational cycle of yeast and human Hsp90 and identifies two key residues that are responsible for the major difference in the catalytic ATPase rate of these Hsp90 proteins. You conclude that the human protein is in a more open state than the human protein and that hHsp90 is more amenable to client protein interaction, enabling interaction with additional clients and co-chaperones.

Major concerns.

1. There is no evidence presented that the human protein is engaged with more client proteins and co-chaperones. Thus, the statement in the abstract that the human protein is more amenable to client protein interaction, enabling interaction with additional clients and co-chaperones should be rephrased. There could be other reasons that the human protein spends more time in a 'more open' state.

We agree and have removed this statement from the abstract.

2. As far as I can tell you do not specify explicitly which Hsp90 isoform you are using. Is the data relating to Hsp90 alpha or beta protein?

This information has been included in the results and method section of the revised version.

Revisions to the main text:

“When we incubated yHsp90 (Hsp82) or hHsp90 (Hsp90β) with the slowly ...”

Revisions to the method section:

“Human Hsp90 (Hsp90β) and mutants were expressed in *E. coli* BL21 cells...”

“yHsp90 (Hsp82) and corresponding mutants were expressed in *E. coli* BL21 cells...”

3. On page 7 of the manuscript, you mention that when CaCl₂ is present in the buffer, both yHsp90 and hHsp90 show substantial decreases in activity. However, you do not refer to any figures that I can see for this result. Please refer the reader to the data.

In the revised manuscript, we now refer to the respective supplementary figure (Fig. S3).

4. NH₄SO₄ had a very different effect on the ATPase activity of yHsp90 and hHsp90. This was attributed to increased closing rates. I would also like to see SEC experiments showing that NH₄SO₄ shifts the hHsp90 to the closed state but not the yHsp90. This would eliminate any other unknown effects the NH₄SO₄ may have on the assay.

This effect is depicted in Fig. 1G, depicting the ratio of open and closed states. The indicated percentages and resulting closing rates were determined from the SEC elution profiles of both yeast and human Hsp90. Notably, when NH₄SO₄ was present in the buffer, complete formation of the closed state was not observed for yeast Hsp90, as indicated by a lower percentage of the closed state compared to the KCl buffers. Conversely, for human Hsp90, a near-complete formation was measured after 70 minutes of incubation.

6. Ala 31 is actually T31 in human Hsp90 beta. This is a highly conserved residue position in Hsp90 and I do not understand why you say the residue is Ala 31.

Thanks for noting this. The typo has been corrected.

7. Similarly yeast Ser109 is Glutamine, not glutamate (Gln 118) in hHsp90 beta. Unfortunately, I could not assess the accuracy of your data without knowing the correct mutations. One other point, is there any evidence that Ser109 is phosphorylated? Could the faster rate of closer be due to that lack of phosphorylation at this position in the yeast protein and also the same argument could be had for T13 in yeast. What effect would phosphorylation at these sites have on the yeast protein?

We corrected these mistakes in the revised version. We checked for reports on Ser109 phosphorylation but did not find any information. In general, the effect of phosphorylations in Hsp90 are hard to predict. We and other groups had studied a number of sites in yeast Hsp90 before and found a range of effects. Most of them affected the allosteric communication of the molecule. This information has been added to the discussion in the revised manuscript.

Minor points

1. In Figure 2 A. Rather than give ASN-107 and Ile-101, the mutations should be labelled as A107N and T101I, so it is clear which are the mutations in the figure.

Has been changed as suggested.

Reviewer #4

The ATP-dependent 90 kDa heat shock proteins (Hsp90s) are essential molecular chaperones that assist folding and maturation of client proteins involved in regulating cell homeostasis, proliferation, differentiation, and cell death. The Hsp90 dimer promotes client folding and maturation via an ATP-hydrolysis-driven conformational cycle. In this cycle, the dimer transitions from an “open” V-shaped conformation, in which the N-terminal nucleotide-binding domains are far apart, to a “closed” conformation where the two N-terminal domains dock onto each other enabling ATP hydrolysis. Yeast is frequently used as model organism to study general cell biological processes believed to be conserved through eukaryotic evolution and relevant for human diseases as well. This is also true for the molecular chaperone Hsp90, for which the yeast protein is investigated in much greater detail than the human homologs and it is believed that the mechanism is conserved between yeast and human Hsp90. However, the yeast and human proteins present a 10-fold difference in ATPase rate for which the molecular cause and consequences are unclear.

Riedl and co-workers investigated in depth the differences between yeast and human Hsp90, analysing conformational changes in the two proteins using size-exclusion chromatography (SEC), analytical ultracentrifugation, single molecule Förster Resonance Energy Transfer (FRET), hydrogen-exchange mass spectrometry and molecular dynamics simulations with wildtype and mutant proteins. The authors studied the opening and closing dynamics of human and yeast Hsp90 in the presence of different nucleotides and different salts of the Hofmeister series. They show that differences in ATPase rate between the two proteins depend on similar but distinct closing mechanisms of the N-terminal lid and the α 1-helix. The authors further investigated the influence of known mutations affecting Hsp90's ATPase activity and closing dynamics. The results presented in the manuscript demonstrate that human Hsp90 is more sensitive to changes in hydrophobicity in both the environment and the protein itself. Recently, the notion that ATP hydrolysis is essential for Hsp90 activity was challenged by reports demonstrating that nucleotide exchange is sufficient for maintaining activity in hydrolysis defective Hsp90. The manuscript explores this new concept by investigating ATP γ S displacement by ATP, ADP or radicicol using SEC and analytical ultracentrifugation as readout for the conformation. An elegant Sortase-based ligation method allowed the authors to fluorescently label the cysteine-rich hHsp90 allowing them to compare the closing dynamics of yeast and human protein also in single-molecule FRET measurements. These measurements revealed different kinetics and closing dynamics between the two proteins. Furthermore, hydrogen exchange experiments on the closed and open Hsp90 revealed that the differences in closing dynamics for human and yeast Hsp90 do not lie only in the N-terminal domain but also involve the middle and C-terminal domains. Finally, molecular dynamic simulations performed on the full-length yeast protein and both yeast and human N-terminal domains elucidated details of the N-terminal rearrangement. Interestingly, the authors were able to reverse some of the differences between the human and yeast proteins in the MD simulations by replacing only two residues (yeast T13A and S109Q) that differentiate the two proteins in the N-terminal domain. The same replacements introduced into yHsp90 reduced the ATPase activity almost to the level of wild type hHsp90. The introduction of the reverse mutations into hHsp90 increased the ATPase activity by 3.5-fold.

In summary, this work provides insights into the evolutionary adaptation of Hsp90 chaperones. It significantly expands our knowledge on the dynamics of the nucleotide-driven conformational changes in Hsp90 and how the ATPase cycle was tuned by single amino acid replacements during evolution. The results presented are exhaustive and state of the art and the conclusions largely justified. Given the widespread use of yeast as model system for protein folding diseases in humans, this study will meet considerable interest in the

chaperone community. There are, however, some points the authors should address before publication.

We thank the reviewer for the expert summary and the encouraging comments.

Major concerns:

1. The manuscript presents many inaccuracies in the formatting and presentation of data. The quality of the scientific work displayed can be undermined by poor presentation. For examples see point 9 and points 5 and 8 in the “minor comments” section. We advise the authors to check the manuscript for potential mistakes that could have escaped this reviewer.

We thank the reviewer for these comments and have changed the points mentioned.

2. As one salt of the Hofmeister series the authors used CaCl₂ in concentrations up to 500 mM in their ATPase assays. Calcium phosphates are notorious for being poorly soluble (solubility limit for Ca₃(PO₄)₂, 64 μM; for CaHPO₄, 316 μM). How soluble is Ca-ATP? At 200 to 500 mM Ca²⁺ ions and 5 mM ATP, Ca-ATP could be precipitating, which could explain the reduced ATPase activity of Hsp90 as the residual ATP concentration in solution were considerably lower and Hsp90 has a rather high K_M for ATP (yHsp90: 350 μM at 37°C, Wegele et al. 2003; hHsp90: 840 μM at 37°C, McLaughlin et al. 2002). The authors need to address this question.

The reviewer raises an interesting point, which may be relevant at high Ca concentrations. We searched the literature for K_D values of Ca and ATP but did not find any. As we cannot resolve this issue at the moment, we added this caveat in the revised manuscript.

3. For the crosslinking experiments presented in Figure S4 it is not clear what the times shown in the figure refer to. If the time corresponds to the incubation in the presence of ATPγS, how did the authors prevent further ATPγS binding to Hsp90 during the 45 minutes DSG crosslinking? If the ATPγS binding kinetics was quenched in one way or another (e.g. by lowering the temperature) the authors should show that this regime prevents binding and dissociation of ATPγS. The authors should also clarify whether the crosslinking reaction was quenched with 10 mM Tris as stated in the figure legend or 200 mM Tris as stated in the methods section.

We have addressed this matter by specifying in the figure legend that the indicated time refers to incubation with ATPγS. Since crosslinking inhibits Hsp90 from undergoing further structural rearrangements, such as closing and opening, subsequent binding or dissociation of ATPγS does not influence the reaction. Based on the concentrations utilized, crosslinking should have reached an equilibrium nearly instantaneously.

Regarding the quenching with Tris, we have updated the information accordingly. The final concentration of Tris was 10 mM, while the stock solution had a concentration of 200 mM.

Changes in the Supplementary Information:

“A: Exemplary SDS-PAGE analysis of crosslinked hHsp90 after different time points of incubation with ATPγS.”

“Subsequently, the reaction was quenched by the addition of 200 mM Tris (yielding a final Tris concentration of 10 mM).”

4. In Figure 2 B-C-D-E it is not clear how many replicates have been performed for each mutant and the ATPase activity of yeast mutant T22I seems to lack any error bars.

We have added the information on replicates as requested. The error bars for T22I are included. They are very small.

5. At the bottom of page 11, the authors refer to “ADP-induced opening”, however, the experiment they refer to in Figure S7 was performed using radicicol to induce Hsp90 opening, and not ADP, making the author's statement inaccurate.

We have changed the wording to „radicicol-induced“.

6. For the smFRET measurement, the length of the incubation with nucleotides is never stated.

We apologize for this omission. For the smFRET experiments, we incubated yHsp90 for 2 hours and hHsp90 for 4 hours. We have now included this information in the figure captions of Figure 4, Figure S10, Figure S11, Figure S12 and Figure S13, and in the Material and Methods section of the smFRET experiments.

7. In their smFRET measurement in Figure 4A, the authors found for yHsp90 in the presence of AMPPNP a FRET efficiency of 0.08. Assuming a Förster radius of 60 Å (value missing in the manuscript; should be provided), these data are consistent with an average inter-dye distance of 88 Å as found in the crystal structure of AMPPNP bound yHsp90 with docked N-terminal domains. In contrast, for hHsp90 FRET efficiencies up to 1 were measured in the apo state and in the presence of ATPγS or AMPPNP. Since in the presence of AMPPNP the N-terminal domains dock onto each other, it is difficult to understand how hHsp90 can exhibit FRET efficiencies up to 1, meaning an inter-dye distance of less than 40 Å. The N-terminal domains would need to rotate around the length axis of Hsp90 to bring the dye molecules closer together. This should be discussed in more detail.

The reviewer is correct. Regarding the Förster Radius, we used an $R_0 = 59 \text{ \AA}$, which we now also mention in the Material and Methods section on *SmFRET measurements and data analysis* in addition to the section of the *Accessible volume calculations*. As the reviewer correctly points out, the observed FRET efficiencies of > 80% indicate that the labeled locations of the NTDs come closer together to within ~ 40 Å. Hence, the domains undock and rotate on the submillisecond timescale. We now mention this in the results section regarding the smFRET experiments as well as in the discussion.

Additions to Main Text

"This indicates that hHsp90 is more dynamic than its yeast counterpart and that the NTDs undock and rotate, which transiently brings the fluorophores into close proximity, and as also supported by our molecular simulations (see below)."

Furthermore, new cgMD data has been included in the revised version to address this point and come to similar conclusions. See our response to Reviewer 1, comment 4.

8. For the HDX experiments, the authors should provide as supplementary information a list of all peptides analysed, their residue numbers and relative exchange under the different conditions used.

As suggested by the reviewer, we have provided additional information on the results of the HDX experiments in the supplementary information. Specifically, we have included figures on the corresponding peptide coverage as well as the relative fractional uptake by residue (Fig. S15 - S18).

9. In the paragraph describing the MD simulations, there is some confusion of residue numbers and identity. For example, on top of page 18, the authors refer to hHsp90 Ala13 and

Glu118 and the respective Thr22 and Ser109 in yeast while two lines further down the authors claim to have mutated T13A/S109Q in yeast mimicking A22 and Q122 of the human protein. Q stands for glutamine not glutamate (Gln not Glu). I hope the molecular modelling experts changed the correct residues.

We apologize for the mislabeling. This has been corrected in the revised version.

10. From the observation that addition of ADP, Al³⁺ and F⁻ ions to Hsp90 does not lead to the closed conformation that authors conclude that “the opening of the dimer occurs during the transition of the γ -phosphate from the tetrahedral conformation into the trigonal bipyramidal pre-hydrolysis state”. Unless there is a crystal structure that shows that ADP·AlF₄ bound to yeast or human Hsp90 mimics the trigonal bipyramidal pre-hydrolysis state of ATP, this conclusion is not justified. The ATP binding pocket of Hsp90 is a composite binding site formed between NTD and MD, two domains that are tethered to each other only by a flexible linker, with the NTD only binding the adenine, ribose and α and β phosphate and the MD contacting the γ phosphate, it is not clear whether ADP and AlF₄ form the ATP pre-hydrolysis mimetic. Just adding ADP and Al³⁺ and F⁻ ions to Hsp90 is no guarantee that the trigonal bipyramidal pre-hydrolysis state of ATP is trapped. The mitochondrial Hsp90 TRAP1 was crystallized with ADP-AlF₄ in the docked state strongly arguing against the claim of the authors (Lavery et al. 2014).

The reviewer’s point is well taken. Indeed, we do not have additional evidence for our conclusion. We rephrased the sentence stating this and added a caveat concerning AlF₄ and the composite ATPase site of Hsp90.

Revisions to the main text

“Therefore, the standard effects of ADP-AlF₄ would be consistent with the notion that the opening of the dimer occurs during the transition of the γ -phosphate from the tetrahedral conformation into the trigonal bipyramidal pre-hydrolysis state. However, currently it cannot be ruled out that, due to the composite ATPase site of Hsp90, ADP-AlF₄ is not the ideal mimetic for the transition state and therefore may not unambiguously report on the role of the transition state during opening.”

Minor comments:

1. The authors state “Heat shock proteins (Hsp) are a highly expressed, ubiquitous class of molecular chaperones”. This is an incorrect statement. Not all heat shock proteins are molecular chaperones, nor are all chaperones heat shock proteins!

The reviewer is of course correct. We changed the order of terms: Molecular chaperones are a highly expressed, ubiquitous class of heat shock proteins.”

2. The authors state “from yeast to human, Hsp90 has undergone evolutionary changes”. Such a statement implies that human Hsp90 descended from yeast, which is, of course, not the case. Both yeast and human Hsp90 descended from a common ancestor some 700 million years ago and evolved in different directions selected by different constraints. It may very well be that human Hsp90 serves a larger diversity of clients than yeast Hsp90 and therefore requires a higher flexibility. The authors should state this more correctly.

Has been changed as suggested.

3. The author should define the error bars (standard deviation or standard error of the mean) in all figure legends of the manuscript and include in the panels an indication which protein, yeast or human Hsp90, was being analysed.

The information has been added as suggested.

4. The authors should clearly state whether they are analysing human Hsp90 α or Hsp90b.

The information has been added as suggested.

5. The authors state that human and yeast Hsp90 have an 80% sequence similarity. Sequence similarity is not a rigorously defined term and the degree of sequence identity should be used instead.

We have changed the text to “sequence identity” and “structural similarity”.

6. In Figure 1A the authors should label the different structural elements to avoid confusion of the reader. In particular the catalytic loop is easily overlooked.

The information has been added as suggested.

7. In supplementary figure S1 the X axes should refer to Svedberg rather than volume.

Has been changed as suggested.

8. In supplementary figure S2 the authors should clarify whether time 0 represents a sample containing protease that is immediately quenched or a protease-free sample.

This point has been clarified in the revised version.

9. For Figure 1G did the author add any ATP γ S or other nucleotides to obtain closure in the Hsp90s? This should be indicated in the figure legend.

This point has been clarified in the revised version.

10. Figure S3 misses the Y axes titles.

The information has been added as suggested.

11. Figure 1 F and S3: For hHsp90 the ATPase activity at 500 mM KCl is higher than the ATPase activity at 200 mM in Fig. 1F but lower in Fig. S3. The x-axis labels seemed to be inverted in Fig. S3. Number of data points and significance should be indicated in all bar graphs.

The information has been added as suggested.

12. In Figure 3 the labelling of the panels is likely wrong with C and E being inverted.

Has been changed as suggested.

13. In supplementary figure S15 it is not clear why structures of full-length Hsp90 are provided next to the panel. What is their relation to understanding the dynamics of the N-terminal domains described in the figure?

The structures have been deleted.

14. In the description of the protein purification protocol, the authors state that a dilution to a final volume of 120 mL is performed but they never state the starting volume. Thus, the

dilution factor is unclear. More clarity would help the reproducibility of the protein purification method described.

The information has been added as suggested.

15. The authors abbreviate the closing rate with “kcat”. This is not logical as “cat” does not relate to “closing” and may lead to confusion as “kcat” is generally used as catalytic constant in steady state Michaelis-Menten type kinetics.

Has been changed to k_{closing} und k_{opening}

16. Panels D and F of Figure 6 are never referred to in the text.

The information has been added as suggested.

17. The reference for Hofmeister does not seem to be correct.

Has been corrected.

References

1. Jussupow, A. & Kaila, V.R.I. Effective Molecular Dynamics from Neural Network-Based Structure Prediction Models. *J Chem Theory Comput* **19**, 1965-1975 (2023).
2. Jussupow, A. et al. Extended conformational states dominate the Hsp90 chaperone dynamics. *J Biol Chem* **298**, 102101 (2022).
3. Souza, P.C.T. et al. Martini 3: a general purpose force field for coarse-grained molecular dynamics. *Nat Methods* **18**, 382-388 (2021).
4. Abraham, M.J. et al. GROMACS: High performance molecular simulations through multi-level parallelism from laptops to supercomputers. *SoftwareX* **1-2**, 19-25 (2015).
5. Tribello, G.A., Bonomi, M., Branduardi, D., Camilloni, C. & Bussi, G. PLUMED 2: New feathers for an old bird. *Computer Physics Communications* **185**, 604-613 (2014).
6. Bonomi, M. et al. Promoting transparency and reproducibility in enhanced molecular simulations. *Nature Methods* **16**, 670-673 (2019).
7. Thomasen, F.E., Pesce, F., Roesgaard, M.A., Tesei, G. & Lindorff-Larsen, K. Improving Martini 3 for Disordered and Multidomain Proteins. *Journal of Chemical Theory and Computation* **18**, 2033-2041 (2022).
8. Jussupow, A. et al. The dynamics of linear polyubiquitin. *Science Advances* **6**(2020).
9. Jumper, J. et al. Highly accurate protein structure prediction with AlphaFold. *Nature* **596**, 583-589 (2021).
10. Pfaendtner, J. & Bonomi, M. Efficient Sampling of High-Dimensional Free-Energy Landscapes with Parallel Bias Metadynamics. *Journal of Chemical Theory and Computation* **11**, 5062-5067 (2015).
11. Raiteri, P., Laio, A., Gervasio, F.L., Micheletti, C. & Parrinello, M. Efficient reconstruction of complex free energy landscapes by multiple walkers metadynamics. *Journal of Physical Chemistry B* **110**, 3533-3539 (2006).
12. Bottaro, S., Bengtson, T. & Lindorff-Larsen, K. Integrating Molecular Simulation and Experimental Data: A Bayesian/Maximum Entropy Reweighting Approach. *Methods Mol Biol* **2112**, 219-240 (2020).

REVIEWERS' COMMENTS

Reviewer #1 (Remarks to the Author):

The authors have done a good job in addressing all major concerns of the Reviewer. The revised manuscript presents a considerable improvement of the original submission and the main results are well described and support the conclusions of this study.

Reviewer #2 (Remarks to the Author):

The authors addressed all the prior concerns. The manuscripts now stands as an important contribution to the Hsp90 molecular chaperone field.

Reviewer #3 (Remarks to the Author):

Dear Johannes and co-authors,

Thank you for revising the suggestions and I am happy to say that I am satisfied with the revisions I suggested.

Reviewer #4 (Remarks to the Author):

The authors have adequately addressed all reviewers' comments and criticism.